# GET: Rethinking Dynamic Graph Learning as Global Event Sequence Generation

## Abstract

The prevailing paradigms in dynamic graph learning, which rely on local neighbor propagation or node-specific memory updates, are fundamentally limited in their ability to model global interaction patterns and to scale inference efficiently. Rethinking these challenges, we propose GET (Global Event Transformer), an *event sequence modeling framework* that reformulates dynamic graph learning as global event sequence modeling. By processing the entire interaction history as a unified stream of events, GET is designed to capture dependencies beyond the local receptive fields of prior methods, while its "encode once, score many" design removes the candidate-wise inference bottleneck of discriminative models. GET flexibly incorporates structural priors through lightweight GNN encoders and memory modules, which provide local structural and temporal cues that the global Transformer then reasons over. Our main focus in this paper is *dynamic link prediction* under standard Temporal Graph Benchmark (TGB) protocols, complemented by preliminary short-horizon generative evaluations on small/medium graphs. Extensive experiments on five large-scale TGB benchmarks and six additional datasets show that GET achieves strong competitive performance while delivering substantially faster inference throughput (up to $21.4\times$ on tgbl-wiki) compared to strong baselines.

## 1 Introduction

Real-world systems, such as social platforms, financial networks, and recommender systems, generate vast streams of time-stamped interactions among entities (Holme & Saramäki, 2012). These processes are naturally modeled as *temporal graphs*, where nodes represent entities and edges denote interaction events annotated with continuous timestamps (Kazemi et al., 2020). A fundamental challenge in such systems is *predicting future interactions* based on historical records, known as *dynamic link prediction*, which underpins numerous applications including recommendation, anomaly detection, and user behavior modeling (Gravina & Bacciu, 2024).

**Limitations of existing methods.** Existing methods (Feng et al., 2023; Liu et al., 2024; Li et al., 2023) for dynamic graph learning primarily follow two dominant paradigms. *Memory-based methods*, such as TGN (Rossi et al., 2020) and DyRep (Trivedi et al., 2019), maintain and update per-node memory states to propagate temporal dynamics. *Structure-based models*, including TGAT (Xu et al., 2020), CAWN (Wang et al., 2021b), and DyGFormer (Yu et al., 2023), instead rely on time-aware neighborhood aggregation to capture local topological evolution. Hybrid approaches like TNCN (Zhang et al., 2024b) and TCL (Wang et al., 2021a) attempt to fuse these ideas by jointly modeling memory and neighborhood structure.

Despite these advances, fundamental limitations remain:

*First*, existing methods suffer from **limited receptive fields**. Memory-based models only update local node states and struggle to propagate global signals efficiently, especially when long-range dependencies span multiple hops or involve disjoint subgraphs. Structure-based models are constrained by sampling neighborhoods within a fixed temporal window, which leads to information loss in sparse or irregular event streams. For instance, TGAT applies time encoding on local neighbors, but fails to model global dependencies across disconnected components.

*Second*, most methods rely on a **discriminative inference paradigm**, where the model scores candidate interactions in a pairwise fashion (e.g., $(s, d)$ or $(s, t, d)$), making the inference cost grow linearly or quadratically with the number of candidates. This severely limits real-time deployment and large-scale application.

**Our approach.** Instead of relying solely on localized propagation, we propose **GET (Global Event Transformer)**, an *event sequence modeling framework* that treats the temporal graph as a single, global stream of interaction events. Architecturally, GET uses an autoregressive Transformer with continuous-time biases (via ALiBi) and a regression head for timestamp prediction, allowing it to encode precise time differences without discretization. In this work, however, we *train and evaluate* GET primarily as a **discriminative link predictor** under the standard TGB protocol: the global event sequence encoder is reused to score candidate destinations efficiently, rather than to deploy a full generative pipeline in the main experiments.

By shifting the focus from per-node histories to a unified event stream, GET directly addresses the aforementioned limitations in receptive field and candidate-wise inference cost. It combines four key advantages:

- **Global dependency modeling**: a Transformer with ALiBi attention captures long-range temporal patterns across all nodes without relying on neighborhood sampling or strictly local memory updates;
- **Efficient inference**: under the standardized TGB evaluation protocol, GET encodes a window of recent events once and reuses this representation to score all candidate destinations in parallel, yielding substantial empirical throughput gains in our experiments;
- **Structure-aware flexibility**: GET incorporates modular structural priors, including GNN-based topological encoders and memory-based temporal context, that can be plugged in as needed;
- **Scalable generalization**: GET performs competitively across diverse dynamic graphs, demonstrating both accuracy and runtime advantages in varied domains.

**Contributions.** Our contributions are threefold: (1) We propose **GET**, an event sequence modeling framework for dynamic graphs that applies a global Transformer over the interaction stream and reuses its representations to support efficient "encode once, score many" link prediction. (2) We design a modular architecture that flexibly incorporates structural priors through a GNN encoder and a TGN-style memory module, clarifying their complementary roles across graphs of different densities. (3) We conduct comprehensive experiments on five large-scale TGB datasets and six additional benchmarks, where GET achieves superior performance on the largest and most challenging benchmarks (e.g., **tgbl-comment**) while remaining highly competitive on other widely studied datasets and delivering substantial empirical inference speedups. While GET's architecture supports autoregressive event generation, this paper focuses primarily on discriminative dynamic link prediction under the standard TGB protocol, with preliminary generative experiments reported in Appendix F.

## 2 RELATED WORK

Dynamic graph modeling (Gastinger et al., 2024; Wang et al., 2024; Yi et al., 2025b; Cornell et al., 2025) has evolved along three major paradigms: memory-based methods, structure-enhanced methods, and Transformer-based global sequence models.

**Memory-Based Methods.** Memory-based methods maintain per-node states to model long-term temporal dependencies (Ji et al., 2024; Cheng et al., 2024; Zhou et al., 2023). Early works like JODIE (Kumar et al., 2019) and DyRep (Trivedi et al., 2019) update node embeddings through recurrent networks or temporal point processes. TGN (Rossi et al., 2020) introduces a message-passing memory mechanism, establishing a standard template for many later models. Lightweight memory variants such as EdgeBank (Poursafaei et al., 2022) replace parameterized updates with edge-level caching for scalability. Despite their strengths in capturing node histories, these models typically rely on local neighbor updates, limiting their global context modeling.

**Structure-Based Methods.** Structure-enhanced methods focus on encoding the neighborhood structure of each node (Lu et al., 2024; Luo et al., 2023; Zhang et al., 2024a). TGAT (Xu et al., 2020) integrates temporal encoding with graph attention to aggregate local neighbors. CAWN (Wang et al., 2021b) captures temporal neighborhood dynamics through random walk patterns. DyGFormer (Yu et al., 2023) and NAT (Luo & Li, 2022) extend this line by combining neighborhood co-occurrence

or multi-hop dictionaries with Transformer layers. However, these methods still operate within localized receptive fields and often scale poorly with large candidate sets.

**Sequence and Transformer-Based Methods.** Inspired by advances in NLP, recent works propose sequence-centric modeling of dynamic graphs (Wang et al., 2025; Chen et al., 2025; 2024; Yi et al., 2025a). For instance, SimpleDyG (Wu et al., 2024) treats the interactions of individual nodes as independent sequences. Recent sequence-based temporal models such as ROLAND (You et al., 2022) and DyGMamba (Ding et al., 2025) further explore state-space and edit-based architectures for dynamic interaction streams, but are mostly evaluated on dynamic graph benchmarks outside the TGB suite and often adopt task setups that are not directly aligned with our evaluation protocol. For reference, we include the recent sequence-based model DyGMamba in our unified small-scale benchmark comparison (Table 2); its original paper evaluates on different temporal graph benchmarks and task setups.

**Key distinction.** Sequence-based temporal models such as SimpleDyG and DyGFormer typically operate on *node-centric* histories, maintaining a separate sequence per node. In contrast, GET models a single *global chronological stream* of all events, enabling direct attention between arbitrary event pairs without routing information through intermediate nodes.

Compared to previous paradigms, GET reformulates dynamic graph modeling as **global event sequence modeling** using Transformer decoders. Unlike discrete-time models (Sankar et al., 2020), which operate on graph snapshots, GET processes the event stream directly in continuous time; its Transformer models the precise, un-discretized sequence. By optionally incorporating GNN and memory modules as structural priors, GET flexibly trades off global receptive field and graph awareness. While GET's architecture is compatible with autoregressive event generation, we evaluate it primarily as a *discriminative* dynamic link predictor on both large-scale TGB benchmarks and widely used smaller interaction datasets. Accordingly, we compare mainly against link-prediction baselines, while Appendix F reports a brief empirical comparison with generation-oriented temporal graph models. Exploring full-fledged generative dynamic graph tasks is left for future work.

## 3 PROBLEM FORMULATION

A continuous-time dynamic graph (CTDG) is defined as $\mathcal{G} = (\mathcal{V}, \mathcal{E})$, where $\mathcal{V}$ is the set of nodes, and $\mathcal{E} = \{x_i = (s_i, d_i, t_i, \mathbf{m}_i)\}_{i=1}^n$ is the set of temporal interaction events. Here, $s_i, d_i \in \mathcal{V}$ are the source and destination nodes, $t_i \in \mathbb{R}^+$ is the timestamp, and $\mathbf{m}_i \in \mathbb{R}^{d_m}$ is an optional message feature. For modeling, we serialize the events in chronological order as $X_{1:n} = (x_1, x_2, \ldots, x_n)$, where $t_1 \leq t_2 \leq \cdots \leq t_n$. This serialization bridges the gap between graph-based dynamics and sequential modeling. Given historical observations $X_{1:n}$, one can model the next event $x_{n+1} = (s_{n+1}, d_{n+1}, t_{n+1})$ via the factorization

$$P(x_{n+1} \mid X_{1:n}) = P(s_{n+1} \mid X_{1:n}) \cdot P(d_{n+1} \mid X_{1:n}, s_{n+1}) \cdot P(t_{n+1} \mid X_{1:n}, s_{n+1}, d_{n+1}). \quad (1)$$

This factorization provides a unified view that can support both link prediction and timestamp modeling by separating "who interacts with whom" from "when the interaction happens". In this paper we instantiate GET mainly as a discriminative model for predicting $d_{n+1}$ (with an auxiliary regression loss for $t_{n+1}$), and use this factorization as a conceptual tool to organize the architecture.

## 4 THE GET FRAMEWORK

GET reframes future link prediction as event-sequence modeling with four modules: **(1) a structure-aware encoder** provides topological and temporal context for nodes; **(2) triplet tokenization** maps events to $(s, d, \text{sep})$; **(3) an ALiBi-enhanced Transformer** models long-range spatio-temporal patterns over the event stream; and **(4) a dual-head decoder** jointly predicts destination nodes and timestamps. Fig. 1 provides an overview of the GET framework and key modules.

### 4.1 STRUCTURE-AWARE CONTEXT MODULES

To provide the model with topological context, we define a structure-aware embedding matrix $\mathbf{H}_{SA} \in \mathbb{R}^{|\mathcal{V}| \times d_{\text{model}}}$. GET supports three variants: in the *Base* version, $\mathbf{H}_{SA}$ is a standard learnable node embedding matrix initialized randomly, which relies purely on the Transformer to learn

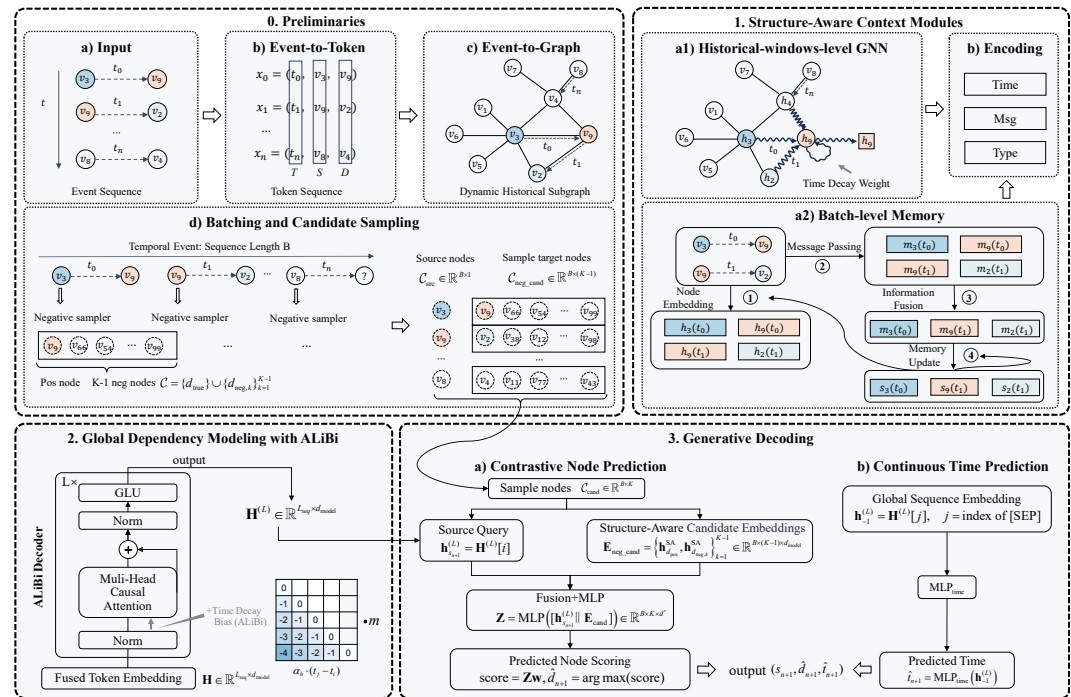

Figure 1: The overview of GET with four stages. (1) **Preprocessing:** The event sequence is tokenized; recent interactions are gathered into a time-decayed computational graph capturing local structure; and mini-batches with negative samples are prepared. (2) **Structure-Aware Context Modules:** Nodes are encoded via a GNN or memory module to obtain embeddings $\mathbf{h}_v$. (3) **Global Dependency Modeling:** The sequence is processed by a multi-layer ALiBi Transformer with time-decay bias to capture global dependencies and yield $\mathbf{H}^{(L)}$. (4) **Generative Decoding:** A dual-head decoder predicts the next node via contrastive matching and the event timestamp via regression.

patterns from the event sequence; in the *GNN-Enhanced* version, $\mathbf{H}_{SA}$ is replaced by the output of a GNN encoder that captures explicit multi-hop structural patterns; in the *Memory-Enhanced* version, $\mathbf{H}_{SA}$ is derived from a memory module that tracks the temporal evolution of node states. The GNN and memory modules serve as optional, plug-in components that inject stronger structural priors into the model. We explicitly adopt a division of labor: the GNN or memory module provides lightweight local structural and temporal cues, while the global Transformer over the event sequence is responsible for long-range temporal and cross-node reasoning. Our claims do not rely on the GNN being able to count complex induced subgraphs; any limitations of standard 1-WL-style message passing are mitigated by the Transformer's ability to attend over the global event history.

**GNN-based Node Embeddings.** To initialize nodes with structural priors, we compute static structure-aware embeddings $\mathbf{H}_{GNN} \in \mathbb{R}^{|\mathcal{V}| \times d_{\text{model}}}$ by applying a single-pass GNN encoder. This encoder operates on a **computational graph** constructed from recent events, allowing it to capture timely structural information without discretizing the event stream. For each edge $(u, v, t_{uv})$, we compute a $k$-dimensional time-decayed edge weight vector $\mathbf{w}_{uv,j}$ using exponential kernels:

$$\mathbf{w}_{uv,j} = \exp\left(-\alpha_j \cdot (t_{\text{current}} - t_{uv})\right), \quad \alpha_j > 0. \tag{2}$$

While this exponential decay is an effective and common heuristic, we acknowledge, in line with prior work (Xu et al., 2020), that it may suppress signals from highly frequent interactions. Exploring more adaptive weighting schemes is a promising direction for future work.

Then, a Graph Attention Network (GAT) (Veličković et al., 2018) is employed due to its ability to incorporate time-decayed edge weights and local neighborhood importance, and we found it to be a robust choice across datasets. In Appendix C.5, we evaluate the robustness of our structural encoder by replacing GAT with GCN and GraphSAGE variants. We observe consistent qualitative trends

across architectures, confirming that GET is not tied to a specific GNN variant. We therefore adopt GAT as the default for its effectiveness in handling weighted graphs. The encoder takes a learnable node embedding matrix $\mathbf{E}_{\text{init}}$ and computes the final structure-aware representations $\mathbf{H}_{GNN}$:

$$\mathbf{H}_{GNN} = \text{GAT}(\mathbf{E}_{\text{init}}, \mathcal{G}_{\text{comp}}, \mathbf{w}_{uv,j}), \quad \mathbf{H}_{SA} := \mathbf{H}_{GNN}. \quad (3)$$

**Memory Module.** To capture temporal dynamics, we employ a TGN-style memory module (Rossi et al., 2020) that maintains a time-evolving hidden state $\mathbf{m}_v \in \mathbb{R}^{d_{\text{model}}}$ for each node. Whenever an event $(s_i, d_i, t_i, \mathbf{m}_i)$ occurs, the state is updated via a GRU using a message from the event context:

$$\mathbf{m}_v^{(t_i)} = \text{GRU}\left(\mathbf{m}_v^{(t_{i-1})}, \ \text{MLP}_{\text{fuse}}\left(\mathbf{m}_u^{(t_{i-1})}, \mathbf{m}_i, \phi(t_i)\right)\right), \quad \mathbf{h}_v^{\text{SA}} := \text{MLP}_{\text{mem}}(\mathbf{m}_v) \quad (4)$$

where $v$ is a participant node (e.g., $s_i$) and $u$ the other (e.g., $d_i$), and $\text{MLP}_{\text{fuse}}$ integrates the other node's memory, the event message $\mathbf{m}_i$, and the temporal encoding $\phi(t_i)$ of the timestamp.

## 4.2 EVENT-TO-TOKEN MAPPING STRATEGY

A key challenge is to serialize asynchronous events into a unified token sequence while preserving temporal and structural information. We map each event $(s_i, d_i, t_i, \mathbf{m}_i)$ into a compact 3-token subsequence $[-1, s_i, d_i]$, where $-1$ denotes a special separator. This design decouples structure (source/destination) from temporal prediction and helps the Transformer recognize event boundaries. Formally, token embeddings are constructed as:

$$\mathbf{h}_p = \left(\mathbf{H}_{SA}[\text{token}_p] + \mathbf{E}_{\text{type}}[\text{type}_p]\right) \odot \text{MLP}_{\text{msg}}(\mathbf{m}_i) \quad (5)$$

where $\odot$ denotes element-wise multiplication, $\mathbf{E}_{\text{type}}$ is the token type embedding (separator/source/destination), and $\text{MLP}_{\text{msg}}$ is a projection for the event message $\mathbf{m}_i$. This unified tokenization enables efficient modeling of long-range dependencies; a sequence of $n$ events yields $L_{\text{seq}} = 3n$ tokens for Transformer modeling. We deliberately avoid introducing the timestamp as a separate token. Doing so would increase the sequence length from $3n$ to $4n$ for $n$ events, which in turn would significantly raise the $O(L_{\text{seq}}^2)$ self-attention cost on long histories. Keeping time as a continuous feature (through ALiBi-style biases and the regression head) instead of a discrete token also maintains a clean separation between "who interacts" (discrete node identities) and "when" (continuous time), which we found to be more stable in practice.

## 4.3 GLOBAL DEPENDENCY MODELING WITH ALiBi

Capturing long-range temporal dependencies efficiently is a key challenge. We therefore integrate **ALiBi** (Attention with Linear Biases) into a multi-layer Transformer decoder to enable global modeling with minimal overhead (Press et al., 2022). Given the token sequence $\mathbf{H} \in \mathbb{R}^{L_{\text{seq}} \times d_{\text{model}}}$ (with $L_{\text{seq}} = 3n$), we project embeddings to queries, keys, and values:

$$[\mathbf{Q}, \mathbf{K}, \mathbf{V}] = \mathbf{H} \cdot \mathbf{W}_{\text{QKV}} \quad (6)$$

and reshape into $H$ heads with $d_h = d_{\text{model}}/H$, then add a head-specific bias to the attention logits:

$$\text{bias}_{h,i,j} = \alpha_h(t_j - t_i) + b_{h,i,j}^{\text{struct}}, \quad \alpha_h = 2^{-c(h+1)/H} \quad (7)$$

Here $t_j - t_i$ is the relative timestamp; $\alpha_h$ is a fixed head slope (we set $c=8$). For each event triplet, all three tokens share timestamp $t_i$. Before use, timestamps are standardized using the training-set mean and standard deviation. The optional $b_{h,i,j}^{\text{struct}}$ can encode historical graph priors, though the temporal term alone is typically effective, which is added to the scaled dot product in attention:

$$\text{Attn}(q_i, k_j, v_j) = \text{Softmax}\left(\frac{q_i \cdot k_j^\top}{\sqrt{d_h}} + \text{bias}_{h,i,j}\right) v_j \quad (8)$$

guiding attention toward temporally local interactions while preserving global context, **without increasing complexity**. Compared to sinusoidal or learned positional encodings, ALiBi enables **parameter-free** continuous-time locality and supports **key-value caching** and **causal masking** for efficient autoregressive generation. The Transformer output $\mathbf{H}^{(L)} \in \mathbb{R}^{L_{\text{seq}} \times d_{\text{model}}}$ feeds the decoder, enabling global event prediction conditioned on long-range temporal context. In preliminary experiments on tgbl-wiki, we compared ALiBi with RoPE-style rotary positional encodings and learned positional embeddings and observed comparable link-prediction performance across all three schemes. We choose ALiBi as the default because it is parameter-free, robust to changes in sequence length and timestamp range, and simple to implement; Appendix C.6 covers these results.

## 4.4 GENERATIVE DECODING

GET adopts an autoregressive dual-head decoder for both discriminative link prediction and short-horizon event generation. The decoder uses two parallel heads: contrastive node classification and continuous-time regression.

**Contrastive Node Prediction.** Given a partially observed event sequence $X_{1:n}$ and the next source node $s_{n+1}$, the model predicts the target node $d_{n+1}$ via conditional contrastive classification. For each step, we construct a candidate set $\mathcal{C} = \{d_{\text{true}}\} \cup \{d_{\text{neg},1}, \ldots, d_{\text{neg},K-1}\}$, where $d_{\text{true}}$ is the ground-truth and $\{d_{\text{neg},i}\}$ are negatives from a pre-defined sampler. For each candidate $v_{\text{cand}} \in \mathcal{C}$, the matching score is computed by concatenating the contextualized source representation $\mathbf{h}_{s_{n+1}}^{(L)}$ with the structure-aware embedding $\mathbf{h}_{v_{\text{cand}}}^{\text{SA}}$, followed by an MLP, where $\|$ denotes vector concatenation:

$$\text{score}(s_{n+1}, v_{\text{cand}}) = \text{MLP}_{\text{node}}\left(\left[\mathbf{h}_{s_{n+1}}^{(L)} \| \mathbf{h}_{v_{\text{cand}}}^{\text{SA}}\right]\right), \tag{9}$$

**Continuous Time Prediction.** In parallel, GET predicts the timestamp $t_{n+1}$ using the sequence-level representation $\mathbf{h}_{-1}^{(L)}$ of a special separator token:

$$\hat{t}_{n+1} = \text{MLP}_{\text{time}}\left(\mathbf{h}_{-1}^{(L)}\right). \tag{10}$$

The predicted $\hat{t}_{n+1}$ is rescaled to the original range using a dataset-specific normalization factor.

**Training Objective and Optimization.** GET is trained with a multi-task objective:

$$\mathcal{L}_{\text{total}} = \mathcal{L}_{\text{node}} + \lambda \cdot \mathcal{L}_{\text{time}} \tag{11}$$

where $\lambda$ balances the node classification (cross-entropy over $\mathcal{C}$) and time regression terms. This combination encourages both accurate link prediction and robust modeling of temporal dynamics. Full details of the objective and optimization are provided in Appendix A. The two heads share the same globally contextualized representation and are trained jointly, which empirically leads to temporally coherent $(d_{n+1}, t_{n+1})$ pairs; Appendix F reports structural and temporal statistics.

## 5 THEORETICAL ANALYSIS

We provide a brief theoretical perspective on GET's modeling capacity relative to local message-passing architectures. We do not claim a novel approximation theorem; instead, we specialize existing results on Transformer universality to the setting of temporal MPNNs.

Consider event sequences of bounded length $L$ with features in a compact subset $\mathcal{B} \subset \mathbb{R}^d$. Let $\mathcal{X}_L$ denote this input space, $\mathcal{F}_{\text{GNN},K}$ the class of functions realizable by $K$-layer temporal MPNNs, and $\mathcal{F}_{\text{GET}}$ the class of GET models. Formal definitions are in Appendix B.1.

**Proposition 1** (Approximation). *For any $f \in \mathcal{F}_{\text{GNN},K}$ and $\varepsilon > 0$, there exists a GET model $g$ such that $\|f - g\|_\infty < \varepsilon$, where $\|\cdot\|_\infty$ is the supremum norm over $\mathcal{X}_L$.*

*Proof sketch.* Any $K$-layer MPNN is a finite composition of continuous operations and thus realizes a continuous function on the compact domain $\mathcal{X}_L$. GET subsumes standard Transformers by setting the structural encoder and memory module to identity-like mappings (for this argument, sinusoidal positional encodings suffice in place of ALiBi). By the universal approximation theorem for Transformers (Yun et al., 2020), the result follows. Full details are in Appendix B.2. □

Beyond approximation, GET can capture dependencies that fixed-depth MPNNs provably cannot:

**Proposition 2** (Separation). *For any $K \geq 1$, there exists a task $h$ and sequences $X, X' \in \mathcal{X}_L$ such that: (i) $h(X) \neq h(X')$; (ii) any $K$-layer MPNN produces identical outputs; (iii) a GET model correctly distinguishes them.*

The construction places the discriminative signal $K+1$ hops from the target, outside the MPNN's receptive field (Xu et al., 2019; Morris et al., 2019; 2020) but within GET's global attention. See Appendix B.3. Memory-based models (e.g., TGN) propagate information along time-ordered event chains, which can also limit long-range modeling. A precise formal characterization is beyond scope; we focus on the well-studied $K$-hop limitations of MPNNs.

# 6 EXPERIMENTS

This section presents a comprehensive evaluation of the proposed **GET** framework on dynamic graph link prediction tasks. The study is structured around four key research questions:

- **Q1 (Effectiveness of the Global Event-Sequence Modeling Paradigm):** Can the proposed global event-sequence modeling approach deliver robust and competitive performance across dynamic graphs of varying scales and characteristics? What are its strengths and limitations?
- **Q2 (Contribution of Structural Modules):** How do the structural modules (GNN and memory) contribute to performance in different scenarios? Do they offer complementary benefits?
- **Q3 (Utility of the Auxiliary Task):** Does the auxiliary timestamp prediction task improve link prediction performance? Is this mechanism broadly applicable across datasets?
- **Q4 (Efficiency and Scalability):** Does GET achieve substantial inference speed advantages over discriminative baselines? Does it scale near-linearly with large candidate sets and long sequences?

These research questions are systematically addressed in the subsequent sections through main performance comparisons, ablation studies, and efficiency analyses, providing multi-dimensional insights into both the empirical effectiveness and practical applicability of the proposed framework.

Table 1: Link prediction performance (Test MRR) on large-scale TGB datasets.

| Model | tgbl-wiki | tgbl-review | tgbl-coin | tgbl-comment | tgbl-flight |
|---|---|---|---|---|---|
| TGN | 0.396 $\pm$0.060 | 0.349 $\pm$0.020 | 0.586 $\pm$0.037 | 0.379 $\pm$0.021 | 0.705 $\pm$0.020 |
| DyGFormer | **0.798** $\pm$**0.004** | 0.224 $\pm$0.015 | 0.752 $\pm$0.004 | 0.670 $\pm$0.001 | NA |
| NAT | 0.749 $\pm$0.010 | 0.341 $\pm$0.020 | NA | NA | NA |
| CAWN | 0.711 $\pm$0.006 | 0.193 $\pm$0.001 | NA | NA | NA |
| GraphMixer | 0.118 $\pm$0.002 | **0.521** $\pm$**0.015** | NA | NA | NA |
| TGAT | 0.141 $\pm$0.007 | 0.355 $\pm$0.012 | NA | NA | NA |
| TCL | 0.207 $\pm$0.025 | 0.193 $\pm$0.009 | NA | NA | NA |
| DyRep | 0.050 $\pm$0.017 | 0.220 $\pm$0.030 | 0.452 $\pm$0.046 | 0.289 $\pm$0.033 | 0.556 $\pm$0.014 |
| EdgeBank (tw) | 0.571 | 0.025 | 0.580 | 0.149 | 0.387 |
| EdgeBank (un) | 0.495 | 0.023 | 0.359 | 0.129 | 0.167 |
| TNCN | 0.718 $\pm$0.001 | 0.377 $\pm$0.010 | **0.762** $\pm$**0.004** | 0.697 $\pm$0.006 | 0.820 $\pm$0.004 |
| GET | 0.752 $\pm$0.012 | 0.373 $\pm$0.006 | 0.757 $\pm$0.021 | **0.732** $\pm$**0.017**[*] | **0.895** $\pm$**0.003** |

[*] Under a specialized setting, GET+Memory can achieve a peak MRR of 0.946 (see Appendix C.4).

## 6.1 EXPERIMENTAL SETUP

**Datasets and Metrics.** We evaluate GET on five large-scale TGB benchmarks (Huang et al., 2023), covering diverse dynamics ranging from long-range sparse interaction streams (e.g., tgbl-wiki, tgbl-comment) to high-volume short-horizon graphs (e.g., tgbl-flight), and six small-scale benchmarks focusing on behavioral dynamics (Poursafaei et al., 2022). Table 4 presents the statistical information of the TGB datasets. Following standard practice, we report Mean Reciprocal Rank (MRR) for large-scale datasets and Average Precision (AP) for small-scale datasets, ensuring fair comparison with prior works.

**Baselines and Implementation Details.** We compare GET against a comprehensive suite of representative continuous-time temporal graph learning models, including: JODIE (Kumar et al., 2019), DyRep (Trivedi et al., 2019), TGAT (Xu et al., 2020), TGN (Rossi et al., 2020), CAWN (Wang et al., 2021b), EdgeBank (Poursafaei et al., 2022),TCL (Wang et al., 2021a), GraphMixer (Cong et al., 2023), NAT (Luo & Li, 2022),PINT (Souza et al., 2022), DyGFormer (Yu et al., 2023), TNCN (Zhang et al., 2024b), and the recent sequence-based model DyGMamba (Ding et al., 2025). Baselines follow official or TGB-standard implementations; TGB scores are from the official leaderboard[1] to ensure consistency with prior works. GET is trained on a single RTX 4090 GPU. Per-dataset configs and all hyperparameters are in Appendix C.

**Negative sampling strategy.** For evaluation on TGB benchmarks, we follow the official protocol and use the pre-generated negative edge sets and evaluation scripts provided by the bench-

___
[1] https://tgb.complexdatalab.com/

mark (Huang et al., 2023). This fixes both the number of negatives per positive and the mixture of historical and random negatives across all methods, ensuring that all models are evaluated on exactly the same candidate sets. During training, we adopt standard uniform random negative sampling with $K = 100$ candidate destinations per positive interaction, consistent with common choices in prior work (e.g., TGAT, TGN) and with the examples distributed with TGB. For the small-scale benchmarks in Table 2, we also follow a random negative sampling protocol, as indicated in the caption, so that all methods share the same evaluation candidates. All reported results are means and standard deviations over multiple runs; the deviations are small and comparable to those of baselines, suggesting that our conclusions are not overly sensitive to negative sampling randomness.

**Evaluation Protocol.** In all result tables, we highlight the best-performing model in **bold** and the second-best in underlined. "NA" indicates that a result was not available for a given model-dataset combination from the original leaderboard.

## 6.2 Main Results (RQ1: Overall Performance Evaluation)

We first assess the performance of GET across five large-scale benchmarks. As shown in Table 1, GET achieves the highest MRR on the two largest datasets, **tgbl-flight** (0.895) and **tgbl-comment** (0.732), both with over 44 million edges. This demonstrates GET's strength in modeling global temporal dependencies and its value for large-scale dynamic graphs. On **tgbl-wiki**, GET attains an MRR of 0.752, trailing DyGFormer (0.798), and remains among the leading models. For **tgbl-coin** (0.757) and **tgbl-review** (0.373), GET's performance is on par with the strong baseline TNCN.

Table 2: Overall results for different baselines under the random negative sampling strategy. All results are obtained by running the official implementations under the unified evaluation protocol.

| Model | MOOC | LastFM | Enron | Can. Parl. | US Legis. | UN Trade |
|---|---|---|---|---|---|---|
| JODIE | 80.04 ±1.77 | 74.83 ± 1.91 | 83.74 ± 0.35 | 64.09 ±0.37 | 66.86 ± 1.29 | 63.67 ± 0.29 |
| DyRep | 81.67 ± 0.36 | 76.01 ±1.55 | 80.97 ± 2.84 | 62.33 ±1.85 | 68.04 ±0.37 | 61.76 ±0.76 |
| TGAT | 85.73 ±0.15 | 75.33 ±0.21 | 69.93 ±0.88 | 65.46 ±0.53 | 61.42 ±2.21 | 61.16 ±0.14 |
| TGN | 89.12 ±1.25 | 78.68 ±2.94 | 84.77 ±0.84 | 65.25 ±1.59 | 69.11 ±0.37 | 63.45 ±1.24 |
| CAWN | 80.59 ±0.20 | 87.84 ±0.06 | 88.66 ±0.40 | 64.91 ±1.57 | 63.59 ±0.77 | 65.29 ±0.15 |
| EdgeBank | 65.73 ±0.11 | 77.22 ±0.96 | 81.72 ±0.46 | 60.91 ±0.38 | 56.03 ±0.56 | 60.87 ±0.08 |
| TCL | 82.06 ±0.18 | 72.48 ±1.45 | 78.68 ±0.51 | 64.12 ±1.82 | 62.06 ±0.45 | 62.20 ±0.18 |
| GraphMixer | 84.58 ±0.58 | 81.59 ±0.45 | 84.15 ±0.21 | 71.64 ±1.56 | 66.38 ±0.94 | 64.83 ±0.86 |
| NAT | 85.12 ±0.76 | 89.57 ±1.29 | 89.80 ±0.85 | 73.55 ±1.70 | 71.48 ±0.62 | 72.84 ±1.18 |
| PINT | 87.97 ±0.69 | 90.61 ±1.21 | 91.13 ±0.43 | 61.81 ±1.05 | 68.79 ±0.94 | 67.79 ±0.50 |
| DyGFormer | 87.32 ±0.36 | 93.45 ±0.09 | 91.73 ±0.26 | 93.99 ±0.53 | 64.63 ±1.82 | 65.86 ±0.85 |
| DyGMamba | 89.03 ±0.06 | **93.71 ±0.15** | 92.10 ±0.11 | 97.53 ±0.88 | 65.37 ±0.49 | 68.11±0.19 |
| GET | **90.75 ±0.28** | 88.39 ±1.35 | **94.81 ±0.17** | **98.97 ±0.46** | **79.44 ±0.33** | **90.85 ±0.51** |

To further evaluate generalization, we also report performance on six commonly used small-scale datasets. For a direct and comprehensive comparison, we evaluate all baseline methods across the entire test set to obtain a single, overall performance score. This approach assesses a model's ability to predict future links involving both previously seen (transductive) and unseen (inductive) nodes, providing a holistic measure of generalization. The results are presented in Table 2. GET achieves the best AP on **MOOC** (90.75), **Enron** (94.81), **Can. Parl.** (98.97), **US Legis.** (79.44), and **UN Trade** (90.85). Notably, GET significantly outperforms DyGMamba on datasets with complex global interactions, such as **US Legis.** (+14.0%) and **UN Trade** (+22.7%). An interesting exception is the **LastFM** dataset, where GET's performance (88.39) is competitive but trails behind node-centric sequence models like DyGMamba (93.71) and DyGFormer (93.45). We hypothesize this is because the LastFM dataset is characterized by long, individual user histories, a scenario where the patching/state-space mechanisms of these models have a distinct advantage. However, on broader dynamic graphs, GET demonstrates superior robustness and generalization.

## 6.3 Ablation Studies [RQ2 & RQ3: What is the impact of each component?]

GET variants include **Base**, **+GNN**, **+Memory**, and **w/o TimeLoss**. To assess the contribution of each component in GET, we conduct ablation studies on five TGB datasets. Results are in Table 3.

**Complementarity of Structure-Aware Modules.** Our ablation study reveals that the GNN and memory modules offer complementary strengths, allowing GET to adapt its structural priors based on graph density. On structurally dense datasets like **tgbl-wiki** and **tgbl-flight** (with densities of $1.91 \times 10^0$ and $2.04 \times 10^1$, respectively), the GNN-enhanced variant is decisively superior, boosting the base model's MRR to 0.752 and 0.895. Conversely, on sparser graphs such as **tgbl-review** (density $4.63 \times 10^{-3}$), the memory-enhanced variant excels, improving the MRR from a base of 0.293 to 0.373. This trend holds for other sparse datasets, as shown in Table 3. These findings suggest that the GNN module is more effective in scenarios with strong structural signals, while the memory module excels in capturing localized, history-driven dynamics, especially under sparse interactions. This explains the performance patterns observed in our main results (Section 6.2): the strong performance on many small-scale datasets highlights the effectiveness of the GET+GNN variant in capturing dense structural patterns. Conversely, GET's value on large-scale graphs like **tgbl-flight** and **tgbl-comment** stems from its global Transformer backbone, which excels at modeling the long-range sequential dependencies prevalent in massive event streams. For a detailed analysis and model insights on the challenging tgbl-review benchmark, see Appendix E.

Table 3: Ablation results of GET on TGB datasets (Test MRR).

| Model Variant | tgbl-wiki | tgbl-review | tgbl-coin | tgbl-comment | tgbl-flight |
|---|---|---|---|---|---|
| GET | $0.439_{\pm 0.009}$ | $0.293_{\pm 0.017}$ | $0.608_{\pm 0.009}$ | $0.365_{\pm 0.013}$ | $0.848_{\pm 0.011}$ |
| GET w/o TimeLoss | $0.408_{\pm 0.005}$ | $0.276_{\pm 0.011}$ | $0.599_{\pm 0.010}$ | $0.337_{\pm 0.014}$ | $0.842_{\pm 0.025}$ |
| GET+Memory | $\underline{0.707}_{\pm 0.008}$ | $\mathbf{0.373}_{\pm \mathbf{0.006}}$ | $\mathbf{0.757}_{\pm \mathbf{0.021}}$ | $\mathbf{0.732}_{\pm \mathbf{0.017}}$ | $\underline{0.856}_{\pm 0.018}$ |
| GET+Memory w/o TimeLoss | $0.662_{\pm 0.002}$ | $\underline{0.356}_{\pm 0.012}$ | $\underline{0.727}_{\pm 0.032}$ | $\underline{0.705}_{\pm 0.010}$ | $0.833_{\pm 0.011}$ |
| GET+GNN | $\mathbf{0.752}_{\pm \mathbf{0.012}}$ | $0.309_{\pm 0.025}$ | $0.648_{\pm 0.014}$ | $0.615_{\pm 0.008}$ | $\mathbf{0.895}_{\pm \mathbf{0.003}}$ |

**Temporal Modeling Capacity of the Base Model.** Even without structural enhancements, the **base GET model** exhibits strong temporal modeling capabilities. On **tgbl-flight**, it reaches an MRR of 0.848, only slightly below that of GET+GNN (0.895) and GET+Memory (0.856), demonstrating that the global Transformer architecture alone can effectively model long-range event dynamics.

**Benefit of Auxiliary Time Prediction.** To evaluate the impact of the auxiliary time prediction task (TimeLoss), we compare models with and without this component. As shown by the comparisons between row 1 vs. 2 and row 3 vs. 4 in Table 3, removing TimeLoss consistently reduces performance across all datasets, with relative MRR drops of 0.7%–7.7%. The most substantial degradations occur on **tgbl-comment** (-7.7%) and **tgbl-wiki** (-7.1%), highlighting the importance of auxiliary time prediction for capturing complex temporal regularities. Unless specified, we set the loss trade-off coefficient to $\lambda = 0.4$. Appendix C.7 reports a sensitivity study over $\lambda \in \{0.2, 0.4, 0.6\}$ on representative datasets, showing stable performance within this range and $\lambda = 0.4$ offers a good trade-off. These results directly address RQ3, showing that the auxiliary timestamp prediction task consistently improves link prediction across datasets.

## 6.4 EFFICIENCY AND SCALABILITY [RQ4: IS GET EFFICIENT AND SCALABLE?]

We conduct a rigorous evaluation of GET's inference efficiency and scalability, a cornerstone of its practical utility. To this end, we selected four large-scale TGB datasets, each chosen to represent a distinct real-world challenge:

- **tgbl-wiki**: A standard, structurally dense graph to establish a baseline.
- **tgbl-review**: An extremely sparse graph with a lot of nodes in information-poor environments.
- **tgbl-coin**: A graph with 1.3M timestamps to test model on long-range sequential dependencies.
- **tgbl-flight**: A graph with 67M of edges concentrated in few timestamps to test raw throughput.

A primary bottleneck for discriminative temporal graph models (e.g., TGN-style architectures) is the recurrent cost of updating node states and aggregating messages before scoring candidates. While the per-candidate scoring is $O(Kd)$ for both TGN and GET, GET encodes a sliding window of $L$ events once and reuses the representation to score all $K$ candidates in parallel. This "encode once, score many" design avoids repeated propagation, yielding up to $21.4\times$ throughput gains on tgbl-wiki under the TGB protocol. We stress that this is a *regime-specific operational advantage* rather than a complexity-theoretic improvement.

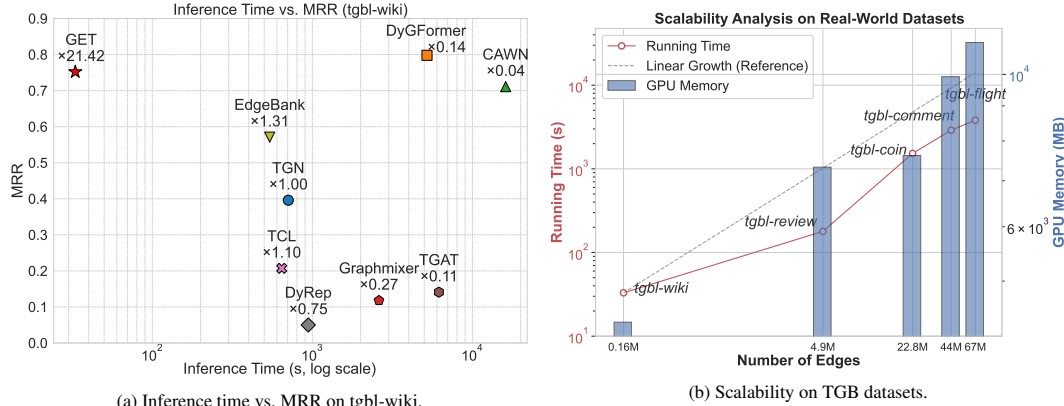

Figure 2: Scalability and efficiency of GET. Panel (a) shows accuracy vs. inference time on tgbl-wiki, and panel (b) shows near-linear growth of inference time and GPU memory with dataset size.

**Favorable trade-off and robust scalability.** GET demonstrates a superior accuracy–speed trade-off, primarily due to its encode-once, score-many event sequence modeling design. As shown in Fig. 2(a), GET is approximately $21\times$ faster than TGN on tgbl-wiki while achieving near-optimal accuracy. This advantage becomes critical on larger graphs: strong baselines like DyGFormer require many hours on tgbl-coin or encounter OOM/timeout on tgbl-flight (Appendix D.1), whereas GET runs efficiently across all scenarios. Fig. 2(b) confirms that GET's runtime and memory scale approximately linearly with edge count.

**Limitations.** Our scalability conclusions are drawn under the standardized TGB evaluation protocol, where inference is dominated by scoring large candidate sets on medium-length windows of recent events. In regimes with extremely long, unsegmented histories (very large $L$) or streaming settings without sliding windows, the quadratic self-attention term can become the dominant factor, and memory-based or linear-time architectures may be preferable. We therefore view GET as a practical complement to existing temporal GNNs and state-space models, particularly well suited to the TGB-style workloads studied in this paper.

Overall, these results demonstrate that GET not only achieves highly competitive accuracy but also provides substantial advantages in inference efficiency and scalability, making it a practically valuable solution for large-scale dynamic graph learning.

## 7 CONCLUSION

In this paper, we revisit the fundamental limitations of existing dynamic graph models, particularly their reliance on localized propagation and pairwise scoring, which restrict their capacity to capture global interaction patterns and to scale inference efficiently. Motivated by these challenges, we propose GET, an *event sequence modeling framework* that treats the temporal graph as a global stream of interaction events and reuses a single Transformer encoding to score many candidate destinations. GET leverages a Transformer-based architecture enhanced by modular GNN and memory components, which provide local structural and temporal cues that the global event encoder then reasons over. Across five large-scale TGB benchmarks and six widely studied smaller interaction datasets, GET delivers strong dynamic link-prediction performance together with substantial empirical inference speedups, especially in regimes where candidate scoring dominates the runtime. We view GET as a practical, modular *complement* to existing temporal GNNs and state-space models, rather than a universal replacement, particularly in candidate-heavy evaluation settings. Beyond the discriminative results that form the main focus of this paper, Appendix F reports preliminary short-horizon generative experiments, indicating that the learned event sequence model can produce structurally and temporally coherent rollouts on small and medium-sized graphs; a full exploration of long-horizon generative objectives and downstream generative tasks is left for future work.

ETHICS STATEMENT

This research is foundational in nature and focuses on the algorithmic aspects of dynamic graph learning. All datasets used are publicly available benchmarks that have been anonymized and widely used by the research community. We do not foresee any direct negative societal impacts or ethical concerns arising from our proposed method.

REPRODUCIBILITY STATEMENT

We are committed to ensuring the full reproducibility of our research. To this end, we provide comprehensive details and resources as follows:

- **Datasets:** All datasets used in our experiments are publicly available. We provide detailed statistics, sources, and any preprocessing steps in Appendix C.1.

- **Methodology:** The architecture and core implementation details of our GET framework, including all model components and formulas, are described in Section 4.

- **Experimental Setup & Hyperparameters:** A complete list of all model hyperparameters, alongside hardware specifications (e.g., NVIDIA RTX 4090 GPUs), key software dependencies (e.g., PyTorch), and the detailed protocol for efficiency measurements are provided in Section 6 and Appendix C.3.

- **Source Code:** The complete source code, including scripts to generate all main results and figures presented in this paper, is included in the supplementary material accompanying this submission. We will also make the repository publicly available upon publication.

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

# A TRAINING ALGORITHM DETAILS

**Training Objective and Optimization.** The GET model is trained using a multi-task objective that jointly optimizes both node prediction and time prediction. The overall loss is formulated as a weighted sum of two components:

$$\mathcal{L}_{\text{total}} = \mathcal{L}_{\text{node}} + \lambda \mathcal{L}_{\text{time}}, \tag{12}$$

where $\mathcal{L}_{\text{node}}$ denotes the node classification loss and $\mathcal{L}_{\text{time}}$ corresponds to the time regression loss, balanced by a hyperparameter $\lambda$.

For the node prediction task, we treat the predicted scores over the candidate set $\mathcal{C}$ as logits and apply the standard cross-entropy loss:

$$\mathcal{L}_{\text{node}} = -\log \frac{\exp(\text{score}(s_{n+1}, d_{\text{true}}))}{\sum_{v \in \mathcal{C}} \exp(\text{score}(s_{n+1}, v))}. \tag{13}$$

This objective encourages the model to assign the highest probability to the true target node among all candidates.

For time prediction, we adopt the Huber loss:

$$\mathcal{L}_{\text{time}}(t, \hat{t}) = \begin{cases} \frac{1}{2}(t - \hat{t})^2, & \text{if } |t - \hat{t}| \leq \delta, \\ \delta|t - \hat{t}| - \frac{1}{2}\delta^2, & \text{otherwise}, \end{cases} \tag{14}$$

where $t$ is the ground-truth timestamp, $\hat{t}$ is the predicted timestamp, and $\delta$ is a threshold parameter, typically set to 1.0. Compared to standard L2 (mean squared error), the Huber loss behaves as L2 for small errors while transitioning to L1 (absolute error) for large deviations, making it more robust to noise and outliers. This robustness is particularly beneficial for real-world dynamic graphs where bursty and irregular events are common. The time prediction loss guides the model to better capture the rhythm and causal dependencies of events, which further enhances link prediction performance.

For optimization, we employ the AdamW optimizer with a cosine annealing learning rate scheduler combined with warm restarts. This combination promotes both stable convergence and improved training efficiency across diverse datasets.

For completeness and to ensure reproducibility, we provide the detailed pseudocode for the training procedure of GET in Algorithm 1 and the autoregressive inference procedure in Algorithm 2.

---

**Algorithm 1** GET Training Procedure

---

**Require:** Training event set $\mathcal{E}_{\text{train}}$, model parameters $\theta$, learning rate $\eta$, batch size $B$, loss weight $\lambda$
**Ensure:** Optimized parameters $\theta^*$
 1: **for** epoch = 1 to max_epochs **do**
 2:     // Step 1: GNN-based structural encoding
 3:     Construct historical graph $G_{\text{hist}}$ from $\mathcal{E}_{\text{train}}$
 4:     $\mathbf{Z}_{\text{GNN}} \leftarrow$ GNN-Enhance$(G_{\text{hist}})$
 5:     **for** each batch $\mathcal{B} \subset \mathcal{E}_{\text{train}}$ **do**
 6:         // Step 2: Event serialization and token embedding
 7:         $X_{\text{token}}, \mathcal{Y}_{\text{node}}, \mathcal{Y}_{\text{time}} \leftarrow$ Tokenize$(\mathcal{B}, \mathbf{Z}_{\text{GNN}})$
 8:         $\mathcal{C}_{\text{nodes}} \leftarrow$ GetCandidates$(\mathcal{B})$
 9:         // Step 3: Forward pass
10:         $\mathbf{H}_{\text{context}} \leftarrow$ Transformer$(X_{\text{token}})$
11:         Score$_{\text{node}} \leftarrow$ DecodeNode$(\mathbf{H}_{\text{context}}, \mathcal{C}_{\text{nodes}}, \mathbf{Z}_{\text{GNN}})$
12:         $\hat{\mathbf{t}}_{\text{pred}} \leftarrow$ DecodeTime$(\mathbf{H}_{\text{context}})$
13:         // Step 4: Loss and parameter update
14:         $\mathcal{L}_{\text{node}} \leftarrow$ CrossEntropy(Score$_{\text{node}}, \mathcal{Y}_{\text{node}})$
15:         $\mathcal{L}_{\text{time}} \leftarrow$ HuberLoss$(\hat{\mathbf{t}}_{\text{pred}}, \mathcal{Y}_{\text{time}})$
16:         $\mathcal{L} \leftarrow \mathcal{L}_{\text{node}} + \lambda \cdot \mathcal{L}_{\text{time}}$
17:         $\theta \leftarrow$ AdamW$(\theta, \nabla_\theta \mathcal{L}, \eta)$
18:     **end for**
19: **end for**
20: $\theta^* \leftarrow \theta$
21: **return** $\theta^*$

---

**Algorithm 2** GET Inference Procedure (Autoregressive Generation)

---

**Require:** Trained model $\theta^*$, historical sequence $X_{\text{hist}}$, number of future events $k$
**Ensure:** Generated future events $X_{\text{pred}}$
 1: $X_{\text{pred}} \leftarrow \emptyset$
 2: // Step 1: Encode structure once
 3: $\mathbf{Z}_{\text{GNN}} \leftarrow$ GNN-Enhance$(X_{\text{hist}})$
 4: **for** $i = 1$ to $k$ **do**
 5:     // Step 2: Tokenize full history
 6:     $X_{\text{token}} \leftarrow$ Tokenize$(X_{\text{hist}}, \mathbf{Z}_{\text{GNN}})$
 7:     $\mathbf{H}_{\text{context}} \leftarrow$ Transformer$(X_{\text{token}})$
 8:     // Step 3: Generate next event
 9:     $s_{\text{new}} \leftarrow$ PredictSource$(\mathbf{H}_{\text{context}})$
10:     $\mathcal{C}_{\text{nodes}} \leftarrow \mathcal{V}$ // candidate set
11:     // avoid overlong line by splitting
12:     Score$_{\text{node}} \leftarrow$
         DecodeNode$(\mathbf{H}_{\text{context}}, s_{\text{new}}, \mathcal{C}_{\text{nodes}}, \mathbf{Z}_{\text{GNN}})$
13:     $d_{\text{new}} \leftarrow \arg\max_{v \in \mathcal{C}_{\text{nodes}}}(\text{Score}_{\text{node}}[v])$
14:     $t_{\text{new}} \leftarrow$ DecodeTime$(\mathbf{H}_{\text{context}})$
15:     $x_{\text{new}} \leftarrow (s_{\text{new}}, d_{\text{new}}, t_{\text{new}})$
16:     $X_{\text{hist}} \leftarrow X_{\text{hist}} \cup \{x_{\text{new}}\}$
17:     $X_{\text{pred}} \leftarrow X_{\text{pred}} \cup \{x_{\text{new}}\}$
18: **end for**
19: **return** $X_{\text{pred}}$

---

# B  THEORETICAL DETAILS

## B.1  DEFINITIONS

**Definition 3** (Event sequence space). Fix a maximum window length $L \in \mathbb{N}$ and feature dimension $d$. Let $\mathcal{X}_L$ denote the set of all event sequences $X = (x_1, \ldots, x_n)$ with $n \leq L$, where each event

$x_i$ has features in a bounded closed subset $\mathcal{B} \subset \mathbb{R}^d$. Under the product topology, $\mathcal{X}_L$ is a compact metric space.

**Definition 4** (Function classes). Let $\mathcal{F}_{\mathrm{GNN},K}$ be the set of all functions $f : \mathcal{X}_L \to \mathbb{R}^m$ realizable by a fixed $K$-layer temporal MPNN architecture (message-passing GNN with $K$ propagation steps). Let $\mathcal{F}_{\mathrm{GET}}$ be the function class of GET models operating on the same input space $\mathcal{X}_L$.

**Definition 5** (Uniform norm). For $f, g : \mathcal{X}_L \to \mathbb{R}^m$, define

$$\|f - g\|_\infty := \sup_{X \in \mathcal{X}_L} \|f(X) - g(X)\|_2.$$

### B.2 Proof of Proposition 1

*Proof.* We argue in three steps. **Step 1: Continuity of temporal MPNNs.** A $K$-layer temporal MPNN on $\mathcal{X}_L$ is constructed by composing: (i) linear transformations on finite-dimensional feature vectors; (ii) pointwise nonlinearities such as ReLU or LeakyReLU; (iii) aggregations (e.g., sum/mean) over finite neighborhoods at each layer. Each of these operations is continuous, and finite compositions of continuous functions on a compact domain remain continuous. Thus any $f \in \mathcal{F}_{\mathrm{GNN},K}$ belongs to $C(\mathcal{X}_L, \mathbb{R}^m)$.

**Step 2: GET subsumes Transformers.** Consider GET with: (i) the GNN encoder set to output a fixed learned embedding for each node, without any message passing; and (ii) the memory module disabled or set to an identity-like mapping that simply forwards its inputs. Under these constraints, the remaining part of GET is exactly a standard Transformer encoder with multi-head self-attention and feed-forward layers, equipped with positional/temporal encodings (ALiBi biases in our implementation, though sinusoidal encodings would also suffice for the theoretical argument). Therefore $\mathcal{F}_{\mathrm{GET}}$ contains the function class of Transformers with positional encodings as a subset.

**Step 3: Universal approximation of Transformers.** By the universal approximation result of Yun et al. (2020), Transformers with positional encodings are universal approximators of continuous sequence-to-sequence functions on compact domains. In particular, for any $f \in C(\mathcal{X}_L, \mathbb{R}^m)$ and any $\varepsilon > 0$, there exists a Transformer $T$ such that $\|f - T\|_\infty < \varepsilon$.

**Conclusion.** Combining Steps 1–3: any $f \in \mathcal{F}_{\mathrm{GNN},K}$ is continuous (Step 1), and there exists a GET model that approximates $f$ arbitrarily closely in the uniform norm (Steps 2–3). This proves Proposition 1. $\square$

### B.3 Proof of Proposition 2

*Proof.* We give an explicit construction.

**Graph and sequences.** Consider a path graph with nodes $v_0, v_1, \ldots, v_{K+1}$. Let $u := v_0$ and $v := v_{K+1}$, so that the shortest-path distance satisfies $\mathrm{dist}(u, v) = K + 1$.

We construct two event sequences $X, X' \in \mathcal{X}_L$ as follows:

- Both sequences contain identical events (with identical features) involving nodes $v_1, \ldots, v_{K+1}$;
- $X$ contains an additional event at node $u$ with a binary feature $z = 1$;
- $X'$ is the same as $X$ except that this event at $u$ has feature $z = 0$.

We define the prediction task as $h(X) := z$, i.e., the goal is to recover the binary feature associated with the distant node $u$.

**MPNNs cannot distinguish $X$ and $X'$.** By the standard receptive-field property of $K$-layer MPNNs (Xu et al., 2019; Morris et al., 2019), the representation $\mathbf{h}_v^{(K)}$ of node $v$ after $K$ message-passing layers depends only on nodes within $K$ hops of $v$. Since $\mathrm{dist}(u, v) = K + 1 > K$, node $u$ lies outside this receptive field. Moreover, the $K$-hop neighborhoods of $v$ in $X$ and $X'$ (including all features and events) are identical by construction. Therefore

$$\mathbf{h}_v^{(K)}(X) = \mathbf{h}_v^{(K)}(X'),$$

and any decoder based on $\mathbf{h}_v^{(K)}$ must produce identical outputs on $X$ and $X'$. Hence no $K$-layer MPNN can implement $h$ on this task.

**GET can distinguish $X$ and $X'$.** In GET, each event (including the one at $u$) is encoded as a token in the global event sequence. The self-attention mechanism allows the token corresponding to the prediction at $v$ to attend directly to the token for the event at $u$. By Proposition 1 and the universal approximation property of Transformers, there exists a parameterization of GET such that: (i) the query at $v$'s position assigns high attention weight to $u$'s event token, effectively extracting the feature $z$; and (ii) a subsequent MLP decoder maps this representation to the correct label $h(X) = z$. For $X'$, the same mechanism yields $h(X') = 0$.

Thus GET can separate $X$ and $X'$ on this task, while $K$-layer MPNNs cannot, completing the proof. □

### B.4 ILLUSTRATIVE EXAMPLE

To make the theoretical advantage more concrete, we consider a specific scenario where nodes A and B are not within $k$ hops in any historical subgraph, nor do their memories interact via a direct chain, but both regularly interact with a third node C (e.g., each Monday). A k-hop GNN's aggregation cannot bridge A and B due to the lack of a sufficiently short path. Similarly, a memory-based model's update chains do not propagate information between A and B, except possibly via C's easily overwritten memory, which struggles to capture long-term periodicity. In contrast, GET's global attention can directly relate any '(A,C)' and '(B,C)' event pairs across the entire history, robustly detecting the periodic co-occurrence regardless of the path.

### B.5 EMPIRICAL INFERENCE THROUGHPUT ANALYSIS

In the standardized TGB evaluation protocol, the dominant cost at test time comes from repeatedly scoring large candidate sets for each query event. All deep temporal models considered in this paper share the same asymptotic per-candidate scoring cost $O(Kd)$ once node representations are available. The observed throughput differences therefore come from how often these representations need to be refreshed and how effectively computation can be batched.

In memory-based architectures such as TGN, each new batch of events triggers recurrent memory updates and, depending on the implementation, repeated neighborhood fetching and message aggregation before candidate scoring. In practice, this leads to substantial overhead on large graphs with long histories.

GET instead performs a single self-attention pass over a sliding window of recent events to obtain a shared contextual representation, and then reuses this representation to score all $K$ candidate destinations in parallel with lightweight MLP heads. Under the TGB workloads we consider, this "encode once, score many" pattern yields the empirical throughput gains reported in Fig. 2 and Table 12, even though the theoretical $O(Kd)$ scoring term is the same. We therefore view GET's efficiency advantage as an *operational* property under this protocol, not as a fundamentally different asymptotic complexity class.

Table 4: Statistics of TGB benchmark datasets. The surprise index (Poursafaei et al., 2022) (i.e., $|E_{\text{test}} \setminus E_{\text{train}}|/|E_{\text{test}}|$) measures the ratio of test edges that are unseen during training; a lower value indicates that memorization-based methods can potentially achieve strong performance. See TGB documentation for more.

| Dataset | tgbl-wiki | tgbl-review | tgbl-coin | tgbl-comment | tgbl-flight |
|---|---|---|---|---|---|
| Nodes (users/items) | 8,227 / 1,000 | 352,636 / 298,590 | 638,486 | 994,790 | 18,143 |
| Edges | 157,474 | 4,873,540 | 22,809,486 | 44,314,507 | 67,169,570 |
| Timestamps | 152,757 | 6,865 | 1,295,720 | 30,998,030 | 1,385 |
| Density (%) | $1.91 \times 10^0$ | $4.63 \times 10^{-3}$ | $5.60 \times 10^{-3}$ | $4.48 \times 10^{-3}$ | $2.04 \times 10^1$ |
| Bipartite | ✓ | ✓ | × | × | × |
| Surprise | 0.108 | 0.987 | 0.120 | 0.823 | 0.024 |
| Weighted (W) | × | ✓ | ✓ | ✓ | × |
| Directed (Di) | ✓ | ✓ | ✓ | ✓ | ✓ |
| Attributed (A) | ✓ | × | × | ✓ | ✓ |

## C  EXPERIMENTAL DETAILS

This appendix provides a comprehensive overview of the datasets, baseline models, and implementation details used in our experimental evaluation, ensuring the reproducibility of our work.

### C.1  DATASETS AND EVALUATION METRICS

We conduct experiments on two categories of datasets to comprehensively evaluate the proposed framework:

**Large-scale datasets**: We adopt five publicly available datasets from the Temporal Graph Benchmark (TGB) as shown in Table. 4, namely tgbl-wiki, tgbl-review, tgbl-coin, tgbl-comment, and tgbl-flight. These datasets are widely used in dynamic graph evaluations and present significant challenges such as structural sparsity, node expansion, and continuous-time dynamics. Following the standard 70%/15%/15% train/validation/test split, we report **Mean Reciprocal Rank (MRR)** as the primary evaluation metric.

**Small-scale datasets**: To further assess generalization capability, we include six publicly available small-scale dynamic graph datasets, namely MOOC, LastFM, Enron, Can. Parl., US Legis., and UN Trade. These datasets, as shown in Table 5 emphasize behavioral dynamics and interaction regularities, making them suitable for evaluating performance under cold-start, weak structural, and non-uniform interaction scenarios. We report **Average Precision (AP)** to facilitate fair comparison with prior works.

A detailed description of each dataset's origin and characteristics is provided below, with full statistics available in the official TGB documentation.

Table 5: Statistics of benchmark dynamic graph datasets. Surprise index as defined in (Huang et al., 2023). "Repeat ratio" and "Density" are computed as in (Huang et al., 2023). "Bipartite": ✓ indicates a bipartite graph; × indicates otherwise. "Domain" is based on the original dataset description.

| Property | MOOC | Enron | Can. Parl. | US Legis. | UN Trade | LastFM |
|---|---|---|---|---|---|---|
| Nodes | 7,144 | 184 | 734 | 225 | 255 | 1,980 |
| Total Edges | 411,749 | 125,235 | 74,478 | 60,396 | 507,497 | 1,293,103 |
| Unique Edges | 178,443 | 3,125 | 51,331 | 26,423 | 36,182 | 154,993 |
| Timestamps | 345,600 | 22,632 | 14 | 12 | 32 | 1,283,614 |
| Duration | 17 months | 3 years | 14 years | 12 congresses | 32 years | 1 month |
| Time Granularity | Unix timestamps | Unix timestamps | years | congresses | years | Unix timestamps |
| Density (%) | 60.2 | 376.0 | 41.6 | 119.9 | 783.5 | 132.0 |
| Bipartite | ✓ | × | ✓ | ✓ | ✓ | ✓ |
| Domain | Interaction | Social | Politics | Politics | Economics | Interaction |
| N&L Feat | – & 4 | – & – | – & 1 | – & 1 | – & 1 | – & - |

### C.2  BASELINE DESCRIPTIONS

To ensure a comprehensive and fair evaluation, we compare GET against a suite of twelve representative baselines, which can be broadly categorized into two major methodological paradigms based on their core mechanism for capturing temporal dynamics.

**Memory-Based Methods.**    These models focus on maintaining an explicit, evolving memory state for each node, which is continuously updated by the event stream to capture historical dependencies.

- **JODIE** (Kumar et al., 2019): Employs coupled Recurrent Neural Networks (RNNs) to learn the embedding trajectories of users and items, capturing their state evolution over time.
- **DyRep** (Trivedi et al., 2019): Models the dynamics of interactions using temporal point processes, focusing on the intensity and association strength between nodes.
- **TGN** (Rossi et al., 2020): A foundational framework that introduces a dedicated memory module for each node, which is updated via a message-passing scheme upon new interactions.
- **EdgeBank** (Poursafaei et al., 2022): A lightweight and efficient non-parametric method that predicts future links based on a cache of recent historical edges between nodes.

**Structure-Based Methods.** These methods focus on encoding the graph's topological structure at or near the time of prediction. They typically operate on sampled temporal subgraphs, historical interaction patterns, or temporal walks to capture structural context.

- **TGAT** (Xu et al., 2020): One of the first models to apply self-attention to dynamic graphs, aggregating features from a node's temporal neighbors with a functional time encoding.
- **CAWN** (Wang et al., 2021b): Captures the structural evolution of a node's neighborhood by sampling and encoding causal anonymous walks.
- **TCL** (Wang et al., 2021a): A contrastive learning framework that models causal relationships by constructing cascading interaction graphs.
- **GraphMixer** (Cong et al., 2023): A simple yet effective model that uses an MLP-Mixer architecture to process a node's historical interaction sequence, challenging the necessity of complex graph structures.
- **NAT** (Luo & Li, 2022): Builds a dictionary of multi-hop neighbors for each node to efficiently capture structural information and updates node states via an RNN.
- **PINT** (Souza et al., 2022): An expressive model that uses an injective message passing scheme and incorporates relative positional features based on temporal walk counting.
- **DyGFormer** (Yu et al., 2023): A powerful Transformer-based model that captures local structure by encoding neighbor co-occurrence as "patches" and processes these patches with a Transformer.
- **TNCN** (Zhang et al., 2024b): A hybrid model that enhances TGN's memory mechanism with features based on multi-hop Neural Common Neighbors, explicitly injecting structural heuristics. We categorize it as structure-based due to its core novelty lying in the structural feature engineering.

To ensure a fair and rigorous comparison, the results for these baseline models were obtained as follows:

- For the **TGB** datasets, all baseline results are directly taken from the official, continuously updated TGB leaderboard to ensure consistency with prior works.
- For the **six small-scale datasets** (MOOC, LastFM, etc.), we reproduced the results by running the experiments ourselves. For most baselines, we utilized the implementations provided in the **DyGLib** library, loading their officially tuned best hyperparameter configurations. For models not included in DyGLib (i.e., NAT and PINT), we used their official public codebases and performed a grid search to find the best hyperparameters.
- For the **inference speed experiments**, results for DyRep, EdgeBank, and TGN were obtained using the official TGB example scripts. Results for all other baselines were generated using the **DyGLib_TGB** repository, loading their optimal configurations to ensure a fair comparison of practical runtime.

### C.3 GET IMPLEMENTATION DETAILS

All experiments for our model (GET) were conducted on a server equipped with an NVIDIA GeForce RTX 4090 GPU (24 GB memory) and Intel Xeon Platinum 8370C 128-core CPUs. All experiments were conducted using PyTorch version 2.1 on a server running Ubuntu 22.04 with CUDA version 12.2. To ensure reproducibility, we provide a detailed list of the key hyperparameters used for training GET and its variants across all TGB datasets. Unless otherwise specified, we use a batch size of 256, a hidden dimension of 256, 4 Transformer layers, and 4 GNN layers (for the GET+GNN variant). The AdamW optimizer is used with an initial learning rate of $4 \times 10^{-4}$ and a Cosine Annealing with Warm Restarts scheduler. All reported results are averaged over five independent runs.

For small-scale datasets (including tgbl-wiki, MOOC, Enron, Can. Parl., US Legis., UN Trade, and LastFM), the evaluation step size is set to 32 to accommodate the smaller number of edges; for all large-scale TGB datasets, the default evaluation step size is 256.

A complete list of hyperparameters is provided in Table 6. These parameters were determined via grid search on the validation set of each respective dataset.

Table 6: Key hyperparameters for GET and its variants on TGB datasets.

| Category | Hyperparameter | Value |
|---|---|---|
| **General Parameters** | Hidden Dimension ($d_{\text{model}}$) | 256 |
| | Batch Size | 256 |
| | Learning Rate (Initial) | 4e-4 |
| | Optimizer | AdamW |
| | LR Scheduler | CosineAnnealingWarmRestarts |
| | Loss Weight ($\lambda$) | 0.4 |
| **Transformer** | Num. Layers | 4 |
| | Num. Heads | 8 |
| | Feed-Forward Dim. | 1024 |
| | Attention Dropout | 0.15 |
| | Residual Dropout | 0.1 |
| **GNN Enhancement** | Num. GNN Layers (GAT) | 4 |
| | GNN Hidden Dimension | 256 |
| | GNN Dropout | 0.1 |
| | Num. Decay Rates ($k$) | 8 |
| **Memory Enhancement** | Memory Dimension | 256 |
| | Message Dimension | 256 |

## C.4 PARAMETER SENSITIVITY ANALYSIS

To evaluate the robustness of GET with respect to architectural design, we conduct a sensitivity analysis on the tgbl-wiki dataset, varying the number of GNN layers and Transformer layers respectively.

All other hyperparameters are fixed to the values listed in Table 7. When varying GNN layers, the Transformer depth is fixed to 3; when varying Transformer layers, the GNN depth is fixed to 4.

Table 7: Fixed hyperparameters used in the sensitivity analysis.

| Hyperparameter | Value |
|---|---|
| Batch size | 1280 |
| Weight decay | $5\times10^{-5}$ |
| Event dropout | 0.0 |
| Graph size | 1024 |
| Step size | 32 |
| Time loss weight | 0.3 |

The results, shown in Figure 3(a) and Figure 3(b), reveal a clear "sweet spot" for both parameters. We observe a significant performance gain from 2 to 3 GNN layers (MRR $0.5625 \rightarrow 0.7400$), underscoring the importance of sufficient structural depth. A similar pattern appears with Transformer layers, peaking at 4 layers (MRR 0.7477) and showing robustness from 2–5 layers. These trends suggest that GET is robust across a range of model depths and does not rely on precise hyperparameter tuning.

**Peak Performance Analysis.** To explore the upper limits of our model's capacity, we conducted a specialized experiment on the challenging **tgbl-comment** dataset with the **GET+Memory** variant under a fine-tuned hyperparameter configuration (see Table 8 for details). This setting achieved a state-of-the-art Test MRR of **0.9460** after the first training epoch. This peak performance demonstrates a classic trade-off between training cost and inference efficiency. The one-time training process was computationally intensive, with a single epoch requiring over **550 hours** of computation on an RTX 4090 GPU. However, critically, the resulting model retained its core advantage of fast inference, completing the entire test set evaluation in approximately **1.5 hours**. Due to the

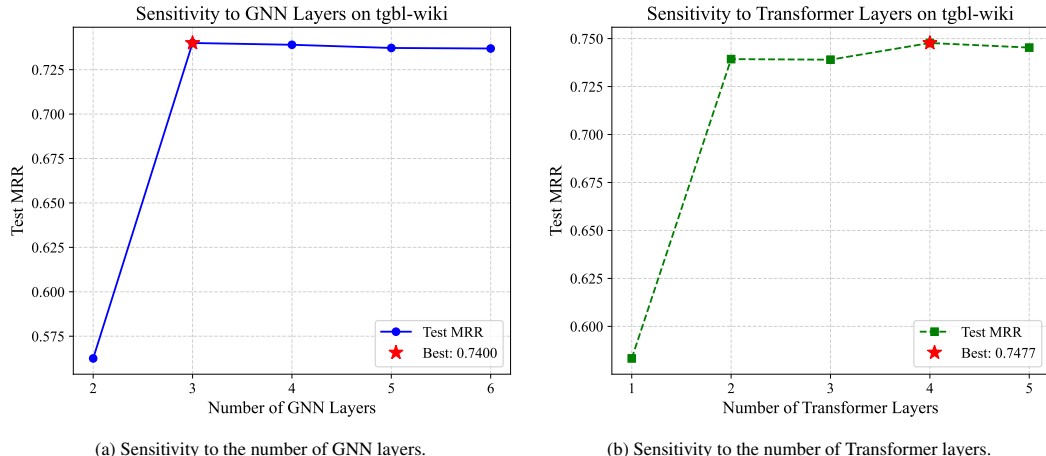

(a) Sensitivity to the number of GNN layers.   (b) Sensitivity to the number of Transformer layers.

Figure 3: Sensitivity analysis on the tgbl-wiki dataset. In panel (a), GNN layers are varied while Transformer layers are fixed at 3. In panel (b), Transformer layers are varied while GNN layers are fixed at 4.

extensive training time, and given that strong performance was achieved early, this result is reported from a single run. This experiment highlights both the high potential of GET's architecture and its consistent inference efficiency, even when trained for maximum accuracy.

Table 8: Hyperparameters for the peak performance experiment on tgbl-comment.

| Hyperparameter | Value |
|---|---|
| step | 128 |
| num_layer | 3 |
| gat_layers | 3 |
| time_loss_weight | 0.5 |
| bs | 1024 |
| prefix | 1024 |
| hiddim | 168 |
| event_dp | 0.15 |
| gat_dropout | 0.15 |

## C.5 ROBUSTNESS TO GNN ENCODER CHOICE

To assess whether GET is sensitive to the specific choice of structural encoder, we evaluated GCN and GraphSAGE variants on the test set.

Table 9: Comparison of GNN encoder variants in GET+GNN (Test MRR).

| Encoder | tgbl-wiki | tgbl-flight |
|---|---|---|
| GCN | $0.738_{\pm 0.005}$ | $0.882_{\pm 0.004}$ |
| GraphSAGE | $0.741_{\pm 0.006}$ | $0.889_{\pm 0.003}$ |
| GAT (default) | $\mathbf{0.752}_{\pm \mathbf{0.012}}$ | $\mathbf{0.895}_{\pm \mathbf{0.003}}$ |

As shown in Table 9, while GAT achieves the best performance, other variants remain competitive, confirming that GET is not strictly tied to a specific GNN architecture.

## C.6 Impact of Positional Encoding Schemes

We also compared several positional / temporal encoding schemes within the Transformer backbone of GET, including learned absolute positional embeddings, RoPE-style rotary encodings, and ALiBi. All variants use the same GET backbone and training protocol; only the positional encoding mechanism is changed. Table 10 shows validation MRR on tgbl-wiki. We observe that all three schemes achieve comparable performance, with differences well within the standard deviations of our main experiments. ALiBi performs slightly better on average and, importantly, is parameter-free and robust to changes in sequence length and timestamp range, which motivates our choice to use it as the default in all main results.

Table 10: Effect of different positional encoding schemes on GET (validation MRR on tgbl-wiki and tgbl-flight).

| Positional encoding | tgbl-wiki | tgbl-flight |
|---|---|---|
| Learned absolute positions | $0.728 _{\pm 0.053}$ | $0.853 _{\pm 0.018}$ |
| RoPE | $0.750 _{\pm 0.005}$ | $0.876 _{\pm 0.041}$ |
| ALiBi (default) | $\mathbf{0.752} _{\pm \mathbf{0.012}}$ | $\mathbf{0.895} _{\pm \mathbf{0.003}}$ |

## C.7 Sensitivity to the trade-off coefficient $\lambda$

We study the sensitivity of GET to the trade-off coefficient $\lambda$ between the node classification loss and the timestamp regression loss. Table 11 reports validation MRR on three representative datasets when varying $\lambda \in \{0.2, 0.4, 0.6\}$. Across this range, the differences are small and comparable to the reported standard deviations; $\lambda = 0.4$ consistently offers a slightly better trade-off, which is why we adopt it as the default choice in the main experiments.

Table 11: Sensitivity of GET to the trade-off coefficient $\lambda$ (Test Metrics).

| Dataset | $\lambda = 0.2$ | $\lambda = 0.4$ (Default) | $\lambda = 0.6$ |
|---|---|---|---|
| tgbl-wiki (MRR) | $0.746 _{\pm 0.008}$ | $\mathbf{0.752} _{\pm \mathbf{0.012}}$ | $0.749 _{\pm 0.010}$ |
| tgbl-comment (MRR) | $0.722 _{\pm 0.015}$ | $\mathbf{0.732} _{\pm \mathbf{0.017}}$ | $0.728 _{\pm 0.014}$ |
| Enron (AP) | $94.65 _{\pm 0.22}$ | $\mathbf{94.81} _{\pm \mathbf{0.17}}$ | $94.70 _{\pm 0.19}$ |

# D Additional Comparisons

## D.1 Inference Efficiency and Memory Usage

**Efficiency Measurement Protocol.** All inference efficiency experiments were conducted on a server equipped with an Intel Xeon Platinum 8370C CPU and NVIDIA RTX 4090 GPUs. To assess practical throughput, we report the end-to-end wall-clock time required for a complete evaluation pass on each dataset. As the inference process for these large-scale benchmarks is deterministic and computationally intensive, a single, uninterrupted run provides a reliable and reproducible measure of real-world performance, which is a standard approach in large-scale systems evaluation. For GPU-accelerated models, measurements were performed with the model as the sole primary task on a dedicated GPU to ensure an isolated and consistent environment. For the CPU-based EdgeBank method, time was measured under similar isolated conditions on the CPU. To maintain a fair comparison of algorithmic efficiency, relative speedup factors (e.g., $21\times$ faster) are calculated exclusively among GPU-accelerated deep learning models.

# E Case Study: The tgbl-review Benchmark

The tgbl-review dataset presents an extreme case for dynamic graph learning, as shown in Table 4: it is exceptionally sparse (density $4.63\times10^{-3}$), with a near-maximal surprise index (0.987), and a

Table 12: Inference time (total wall-clock time in H:MM:SS format) on large-scale TGB datasets with diverse scalability challenges. GET is the only strong baseline that runs robustly across all scenarios.

| Model | tgbl-wiki | tgbl-review | tgbl-coin | tgbl-flight |
|---|---|---|---|---|
| EdgeBank[*] | 0:09:01 | 1:41:06 | 32:16:27 | 245:49:35 |
| TGN | 0:11:47 | 0:58:30 | 5:43:51 | 1:40:56 |
| DyRep | 0:15:42 | 0:58:26 | 5:36:10 | 1:41:26 |
| DyGFormer | 1:26:39 | 4:50:15 | 7:46:44 | OOM/OOT[†] |
| TCL | 0:10:43 | 0:28:38 | OOM/OOT | OOM/OOT |
| GraphMixer | 0:43:25 | 1:31:08 | OOM/OOT | OOM/OOT |
| TGAT | 1:42:52 | 5:08:58 | OOM/OOT | OOM/OOT |
| CAWN | 4:28:49 | 8:47:02 | OOM/OOT | OOM/OOT |
| **GET+GNN** | **0:00:33** | **0:02:59** | **0:25:28** | **1:03:39** |

[*] Run on CPU; time is wall-clock time.
[†] OOM/OOT (Out of Memory/Out of Time) indicates that the run failed due to resource constraints. NA indicates that the data was not provided for this model.

minuscule repeat ratio (0.19%). In this "information vacuum," most test edges are one-off, previously unseen interactions, rendering both local structural signals and global sequential patterns exceptionally weak.

This unique scenario exposes the inductive biases and limitations of different model families:

**(1) Structure-agnostic models excel when there is little structure to exploit.** The best-performing model, GraphMixer (MRR $\sim$0.52), leverages a simple MLP-based design. By not relying on neighborhood aggregation, it avoids overfitting to noisy or non-existent structures, learning a robust mapping from shallow histories.

**(2) Memory modules remain valuable in sparse settings.** TGN (MRR $\sim$0.38) benefits from its node-level memory, which can aggregate the few available interactions of each entity over time. This mechanism remains effective even when the graph structure is fragmented or the node is isolated.

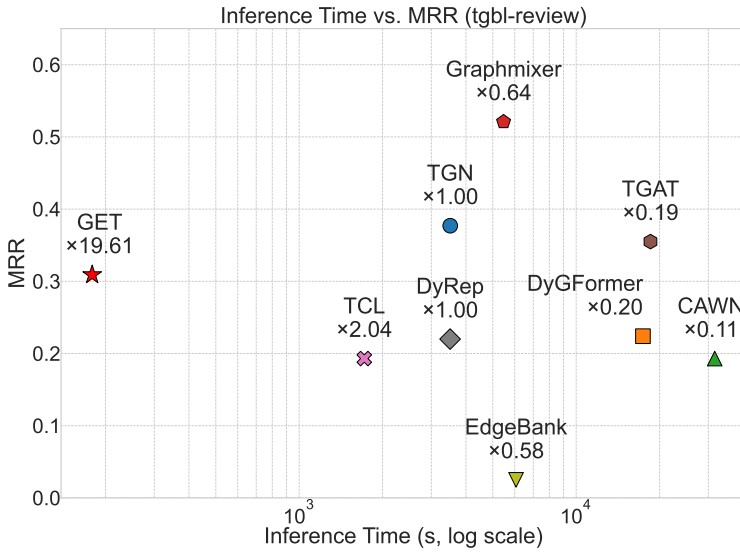

Figure 4: Inference Time vs. MRR on tgbl-review. GET matches or exceeds the accuracy of strong baselines while delivering orders-of-magnitude faster inference.

**(3) GET balances competitive performance with strong efficiency.** Our GET model achieves a respectable MRR ($\sim$0.31), while delivering up to **19.6$\times$ faster inference** compared to memory-based models like TGN, as shown in Figure 4. This substantial gain stems from GET's generative design, which avoids per-candidate scoring and instead generates target nodes in a single forward pass. While GET's global sequence modeling advantage is somewhat limited on tgbl-review—where sequences are short and highly disconnected—it still exhibits strong generalization under sparse supervision. Overall, GET offers a compelling efficiency-accuracy trade-off that is highly suitable for large-scale deployment scenarios.

**(4) Limitations of structural enhancement.** Our ablation study shows that GNN enhancements bring little benefit in tgbl-review: with high sparsity and novelty, neighborhood aggregation cannot provide useful predictive information and may even propagate noise. Conversely, incorporating a memory module (GET+Memory) substantially boosts performance, highlighting the promise of hybrid global-local models for such challenging regimes.

The tgbl-review benchmark vividly demonstrates that (i) strong structural assumptions may hurt in sparse, surprise-rich graphs, (ii) entity memory is essential, and (iii) GET offers an attractive compromise between generalization and efficiency. These findings motivate future research on more adaptive, data-aware structural modules.

# F EVALUATION ON GENERATIVE TEMPORAL GRAPH MODELING

Table 13: **Comparison of GET against multiple baselines** on synthetic dynamic graph generation. Each dataset block reports statistics of the original graph (first row, labeled Median), followed by deviations for generated graphs by DYMOND, TagGen, TIGGER, and our proposed GET (last row per dataset). Lower values indicate better performance.

| Dataset | $|\mathcal{V}|$ | $|\mathcal{E}|$ | T | Method | Time(s) | % Edge overlap | Mean degree | Triangle count | PLE | NC | Global CF |
|---|---|---|---|---|---|---|---|---|---|---|---|
| Wiki-Small | 1.6K | 2.9K | 50 | Median | - | - | 1.1064 | 0.0 | 16.4626 | 44.0 | **0.0** |
| | | | | DYMOND | 69120 | **0.0** | 0.2424 | **0.0** | 11.426 | 32.0 | **0.0** |
| | | | | TagGen | 1800 | 87.169 | 0.0674 | **0.0** | 5.6519 | 0.0 | **0.0** |
| | | | | TIGGER | 14 | 1.2821 | **0.0352** | **0.0** | **4.5005** | 4.0 | **0.0** |
| | | | | GET | **4.01** | 0.03 | 0.8101 | **0.0** | - | 15.0 | **0.0** |
| UC Irvine | 1.8K | 33K | 194 | Median | - | - | 1.5714 | 0.0 | 4.634 | 14.0 | 0.0 |
| | | | | TagGen | 12,480 | 79.356 | 0.1806 | **0.0** | 0.7732 | **1.0** | **0.0** |
| | | | | TIGGER | **125** | 25.0 | **0.076** | **0.0** | **0.4135** | 3.0 | **0.0** |
| | | | | GET | 768.2 | **0.01** | 0.2634 | **0.0** | 2.1994 | 45.0 | **0.0** |
| Bitcoin | 3.7K | 24K | 191 | Median | - | - | 1.8443 | 2.0 | 3.5607 | 13.0 | 0.0102 |
| | | | | TagGen | 18579 | 80.0 | 0.2311 | **0.0** | 0.5207 | **1.0** | 0.0016 |
| | | | | TIGGER | **128** | 24.294 | **0.1217** | 1.0 | **0.294** | 3.0 | 0.0078 |
| | | | | GET | 776.27 | **0.00** | 1.0276 | 3.0 | 5.3285 | 523.0 | **0.0001** |

The main body of this paper focuses on discriminative dynamic link prediction. In this appendix, we provide preliminary experiments that probe whether the event sequence model learned by GET can also be used to generate short-horizon temporal interaction sequences on small and medium-sized graphs.

Our goal here is not to introduce a full-fledged temporal graph generator, but rather to examine whether the autoregressive decoder produces structurally and temporally coherent rollouts when applied in a simple generation protocol. To this end, we follow the evaluation setup of TIGGER (Gupta et al., 2022), comparing GET to specialized temporal graph generation baselines such as DYMOND, TagGen, and TIGGER on standard benchmarks.

## F.1 EXPERIMENTAL SETUP

We adopt the experimental setup from TIGGER (Gupta et al., 2022), where each model is trained on the entire temporal interaction graph and tasked with generating a synthetic graph of identical size (matching the number of nodes and interactions in the original dataset). We evaluate GET against the following generative baselines:

- **Median**: Computes the median values of graph statistics from snapshots of the original graph, serving as a statistical reference (ground-truth baseline) for evaluating the fidelity of generated graphs.

- **DYMOND** (Zeno et al., 2021): A graphlet-based generative model with cubic computational complexity.
- **TagGen** (Zhou et al., 2020): An optimization-driven approach employing heuristic-based path sampling.
- **TIGGER** (Gupta et al., 2022): A state-of-the-art neural generative model employing autoregressive interaction sampling.

Evaluations are conducted on three diverse real-world datasets, including messaging interactions (**UC Irvine** (Kunegis, 2013)), financial transactions (**Bitcoin** (Kumar et al., 2016)), online discussions (**Reddit** (Leskovec & Krevl, 2014)), and collaborative editing patterns **Wiki-Small** (Leskovec & Krevl, 2014)). Dataset details are provided in Table 13. Model performance is evaluated across three primary aspects:

- **Efficiency**: Runtime required to generate synthetic graphs.
- **Novelty**: Percentage of edge overlap with the original graph (lower is better).
- **Fidelity**: Median absolute errors across timestamps for essential graph statistics:
    - Mean degree (**Mean Degree**)
    - Triangle count (**Triangle Count**)
    - Power-law exponent (**PLE**)
    - Number of components (**NC**)

Lower median absolute error indicates better fidelity to the original graph characteristics.

### F.2   RESULTS AND ANALYSIS

Table 13 summarizes the comparative performance of GET and baseline methods. Key observations are summarized as follows.

Regarding **Efficiency**, while TIGGER is faster on these smaller datasets, GET's runtime remains practical and scales better than non-neural baselines like DYMOND and TagGen.

For **Novelty (Edge Overlap)**, GET excels by maintaining near-zero overlap, indicating it generates genuinely new interactions rather than memorizing training data.

Finally, concerning **Graph Fidelity**, GET shows a trade-off. While it struggles to match TIGGER on global statistics like PLE and NC, it accurately models local structures like triangle counts. This suggests GET prioritizes generating novel events with correct local patterns, highlighting a different generative strength compared to baselines.

Overall, these results suggest that GET can serve as a useful *building block* for temporal graph generation, balancing novelty and fidelity while remaining reasonably efficient on the settings we consider. We emphasize that these experiments are preliminary and short-horizon in nature; a more systematic study of long-horizon generation, alternative decoding schemes, and downstream generative applications is an interesting direction for future work.

## LLM USAGE STATEMENT

In this work, we utilized Large Language Models (LLMs) as a tool for language refinement, proofreading, and stylistic adjustments during the writing process. All core ideas, experimental designs, and technical contributions were developed solely by the authors. We take full responsibility for the content and conclusions presented in this paper.

