# OpenReview forum: "GET: Rethinking Dynamic Graph Learning as Global Event Sequence Generation"
_ICLR.cc/2026/Conference — Submitted to ICLR 2026_

### Official Review · Reviewer_D46p · 2025-10-27

**Soundness:** 3
**Presentation:** 2
**Contribution:** 3
**Rating:** 4
**Confidence:** 4

**Summary:**

This paper proposes GET (Global Event Transformer), a generative framework that reformulates dynamic graph learning as global event sequence generation. Instead of performing local message passing or node-level memory updates, GET models the entire temporal graph as a unified chronological sequence processed by a Transformer with ALiBi attention. It can optionally incorporate structural priors via lightweight GNN encoders and memory modules. Experiments on five large-scale TGB benchmarks and six smaller datasets show competitive accuracy and substantial inference speedups.

**Strengths:**

Novelty: Reformulating dynamic link prediction as global event sequence generation addresses a real limitation of local propagation methods. The motivation for moving beyond k-hop GNN receptive fields and pairwise scoring bottlenecks is well-articulated and grounded in practical deployment challenges.

Motivation and Analysis: The architectural design consists global sequence modeling (Transformer), structural encoding (GNN), and temporal dynamics (memory), enabling an interesting synergy among them. The ablation studies (Table 3) show complementarity between GNN (dense graphs) and memory (sparse graphs) modules.

Empirical Results: Strong performance on largest datasets (tgbl-flight: 0.895, tgbl-comment: 0.732) with substantial inference speedups validates the practical value of the event sequence generation.

**Weaknesses:**

Motivation:

Despite framing the problem as "event sequence generation," the paper focuses exclusively on discriminative link prediction. No experiments demonstrate multi-step autoregressive generation, long-term forecasting, or joint modeling of multiple future events. They are the core advantages of generative models. The generative evaluation in Appendix E is preliminary and shows mixed results (struggles with global statistics). So I'm not full convinced on the paradigm shift part.

Theoretical analysis:

Theorem 1 (Appendix B) claims FGNN-k ⊂ FGET and FMemory ⊂ FGET but provides only an informal proof sketch. The argument relies on attention masking simulating k-hop aggregation, but this ignores fundamental differences: GNNs use learned message-passing functions specific to graph structure, while Transformers use generic self-attention. The "periodic co-occurrence" counter-example is vague and not formalized. A rigorous proof (e.g., via approximation theory or simulation arguments with explicit constructions) or toning down the claims would improve credibility.


Scalability analysis:

- Largest graph (tgbl-flight) has 67M edges but only 1,385 unique timestamps, meaning events are highly concentrated in time rather than truly long-sequence scenarios. To claim "long-range dependencies", I think it would be appropriate to evaluate the model on some anti money laundering transaction networks such as from IBM.
- While the 21× speedup over TGN is impressive, comparisons are not fully fair: (i) inference time includes both encoding and scoring, but encoding cost (O(L²)) can dominate for long sequences, (ii) no analysis of how speedup varies with K (candidate set size) is shown beyond the theoretical O(K) vs. O(K·Tpropagate) claim, (iii) batch size and hardware utilization differences across methods are not controlled, (iv) no comparison with approximate nearest-neighbor methods or other efficient retrieval techniques that could accelerate discriminative methods.

Baselines:

I think some sequence-based temporal graph methods should be compared with (and referenced):
- SimpleDyG [Wu et al., WWW 2024]
- ROLAND [You et al., KDD 2022]
- DTFormer [Chen et al., CIKM 2024]
- DyGMamba [Ding et al., TMLR 2025]
- TGEditor [Zhang et al, ICAIF 2023]
Without these baselines, it's hard to determine whether GET's advantages stem from global sequence modeling specifically or simply from using Transformers effectively. Authors does not justify why the event sequence paradigm is necessary when other Transformer-based temporal graph methods exist.

Alternative global modeling architectures:
- The paper focuses exclusively on the event sequence paradigm but does not compare to other approaches for capturing global information:
  - Graph Transformers with structural encodings (GraphGPS [Rampášek et al., NeurIPS 2022], Exphormer [Rampášek et al., ICML 2023])
  - Attention-based architectures that could process entire graph histories.
  - Memory-augmented Transformers that aggregate global context differently.
- This makes it unclear whether the event sequence representation is uniquely effective or just one of several viable global architectures.

GNN evaluation:
- Only GAT is tested for the GNN component. It is important to see how different structural encoding methods affecting the performance.
- Only TGN-style memory is tested. Other memory mechanisms might interact differently with the Transformer backbone.
- No ablation of Transformer architecture choices (decoder-only vs. encoder-decoder, number of heads, FFN dimension).

**Questions:**

See weakness

---

> ### Author Response · Authors · 2025-11-20
>
> We thank the reviewer for the balanced and insightful comments, and for recognizing the motivation and empirical strengths of GET. We address your main concerns below.
>
> **(1) Generative formulation versus discriminative evaluation**
>
> We agree that the original framing over-emphasized the “generative” aspect relative to the actual experiments, which are all discriminative link prediction tasks. Architecturally, however, GET remains an autoregressive next-event model over a global event sequence, jointly modeling destinations and timestamps; this generative design is precisely what allows it to capture long-range temporal dependencies. To be compatible with the standard evaluation setup of TGB and prior dynamic graph work, we follow the community convention and train with a contrastive classification loss and report link prediction metrics on fixed candidate sets.
>
> In the **revised manuscript**, we explicitly describe GET as an **“event sequence modeling framework”** and clarify that its generative capabilities (e.g., multi-step rollout, temporal graph synthesis) are explored only preliminarily in the **Appendix “Evaluation on Generative Temporal Graph Modeling”**. We have removed wording suggesting a paradigm shift in generative dynamic graph modeling and instead emphasize the **practical benefits** of global sequence modeling for link prediction and efficiency.
>
> **(2) Theoretical analysis**
>
> We agree that the previous expressiveness statement was heuristic rather than rigorous. Following your suggestion, we have toned down the claims by renaming “Theorem 1” to an **“Informal Proposition”** and rewriting the appendix to clearly state that it provides **intuition only**. The central point we keep is that global attention over the event stream can directly couple interactions that fixed-hop GNNs must propagate through intermediate nodes; we no longer present this as a formal approximation result. A proper theoretical treatment of expressiveness is explicitly flagged as **future work**.
>
>
> **(3) Scalability analysis and regimes**
>
> We revised Section “Experiments” to make our target regime explicit: the TGB link prediction
> benchmarks focus on moderate event windows (hundreds to low thousands of events) and relatively large candidate sets. In this setting, inference time is dominated by scoring many candidates under standardized negative sampling.
>
> GET’s “encode once, score many” design is particularly well suited to this workload: it amortizes the cost of encoding the event history over all candidates, and we empirically observe up to 21.4× higher query throughput on tgbl-wiki, with similar trends on other TGB datasets. This is not merely an implementation trick but a direct consequence of avoiding repeated memory fetching and neighborhood propagation in the scoring loop. At the same time, we now explicitly acknowledge in the paper that for extremely long unsegmented histories (without sliding windows) the quadratic self-attention term would become dominant and memory-based or linear-time architectures may be preferable. We view GET as a practical complement to these models in candidate-heavy TGB-like regimes, rather than as a universal replacement.

---

> ### Author Response · Authors · 2025-11-20
>
> **(4) Additional baselines and sequence-based methods**
>
> We thank the reviewer for highlighting concurrent sequence-based works. To address this, **we have successfully reproduced DyGMamba** on the six small-scale datasets during the rebuttal period and included it directly in the revised **Table 2** (main text). We want to highlight the DyGMamba comparison
> that is now included in Table 2 of the revised PDF:
>
> | Dataset    | DyGMamba | GET (Ours) |
> |------------|----------|------------|
> | MOOC       | 89.03    | **90.75**  |
> | LastFM     | **93.71**| 88.39      |
> | Enron      | 92.10    | **94.81**  |
> | Can. Parl. | 97.53    | **98.97**  |
> | US Legis.  | 65.37    | **79.44**  |
> | UN Trade   | 68.11    | **90.85**  |
>
> GET outperforms DyGMamba on 5/6 datasets. The gains are especially large on
> datasets with complex global patterns (US Legis. +14%, UN Trade +23%).
>
> **Results analysis.** As shown in the updated Table 2, GET outperforms DyGMamba on **5 out of 6 datasets**:
> - **US Legis. (+14.0%) & UN Trade (+22.7%)**: GET achieves large gains. These datasets contain complex global interaction patterns that are not dominated by single long user histories, where we find GET’s global event modeling particularly effective.
> - **LastFM (-5.3%)**: DyGMamba achieves the best performance (93.71), slightly outperforming DyGFormer (93.45) and GET (88.39). This is consistent with our hypothesis that node-centric sequence models are especially strong on user–item logs with long individual histories.
> -  **Overall:** This comparison reinforces our claim: while node-centric models (DyGMamba/DyGFormer) are specialized for user modeling, **GET provides a more robust solution for general dynamic graphs**, especially those with global dependencies. This demonstrates that GET's architectural choice to model the global event stream is not just efficient, but also effective at capturing complex interaction patterns. While gains on saturated benchmarks such as tgbl-wiki and LastFM are modest, the large margins on US Legis. and UN Trade (+14.0 and +22.7 AP, respectively) show that global event sequence modeling brings substantial benefits exactly in the complex, long-range interaction regimes that motivated our design.
>
> *Note:* For the large-scale TGB benchmarks, DyGMamba is not yet integrated into the official TGB pipeline, which relies on pre-generated negative edge sets and standardized evaluation scripts. Adapting DyGMamba’s data pipeline to this setting requires substantial engineering effort; due to the limited rebuttal time, we therefore focused our additions on the small-scale benchmarks and on clarifying GET’s throughput advantages on TGB datasets.
>
>
>
> **(5) Architectural ablations**
>
> We agree that ablations are crucial for understanding the role of structural priors, memory, and transformer design choices. In the current paper, **Table 3** already reports ablations showing that the GNN encoder and memory module are complementary, particularly across graphs of different densities. The **depth sensitivity plots** in the appendix show that shallow configurations (2–4 GNN layers and 2–4 Transformer layers) achieve a good accuracy–efficiency trade-off, while deeper models provide only marginal gains at higher cost. We have updated the main text to highlight these results and refer to the appendix for additional ablations (e.g., alternative GNN architectures).

---

> > ### Comment · Reviewer_D46p · 2025-11-26
> >
> > Thank you for your response. However my questions have been only partially addressed, especially on the baselines and references. My concerns remain valid and I will keep the current scores.

---

> ### Author Response · Authors · 2025-12-03
>
> Thank you for your thoughtful review. In response to your comments:
>
> 1. We added DyGMamba as a strong baseline on six small-scale datasets; GET performs best on 5/6 datasets, especially on US Legis. and UN Trade (see Table 2 in the revised paper).
>
> 2. We rewrote the theory section into a modest Proposition built directly on existing Transformer/GNN theory, together with a simple separation example. We no longer present this as a new theoretical result, but as architectural intuition with formal backing.
>
> 3. We expanded the ablation discussion to better justify the design choices and hyperparameter settings.
>
> We hope these changes help clarify the contribution.

---

### Official Review · Reviewer_Zw8m · 2025-10-30

**Soundness:** 3
**Presentation:** 3
**Contribution:** 2
**Rating:** 4
**Confidence:** 3

**Summary:**

This paper proposes the Global Event Transformer (GET), a generative framework that fundamentally reframes dynamic link prediction as global event sequence generation. By processing the entire graph history as a single sequence and leveraging Transformer with ALiBi for continuous-time modeling, GET aims to overcome the limited receptive fields and inference scalability issues inherent in prior local/discriminative methods. The architecture integrates modular structural priors. Experiments show GET achieves competitive accuracy while delivering substantial speedup in inference, supporting its utility for large-scale dynamic graphs.

**Strengths:**

1.	This paper provides a fundamental paradigm shift for scalability. The shift from a multiplicative ($O(K \cdot T_{\text{propagate}})$) discriminative inference to an additive ($O(L^2 d + Kd)$) generative paradigm effectively resolves the major bottleneck for real-time applications with massive candidate sets.

2.	GET uniquely combines global sequence modeling with continuous time modeling, enabling the capture of long-range dependencies and non-local patterns previously inaccessible to local or memory-based methods.

3.	The framework's modular design allows for flexible injection of structural priors via GNN or TGN-style memory modules, demonstrating complementary strengths tailored to graph density.

**Weaknesses:**

1.	The GNN encoder (Eq. 2) relies on a computational graph from recent events (L155). Given GET's claim to model global, long-range dependencies, relying on a locally constructed graph for structural priors seems contradictory to the global modeling objective.

2.	The dual-head decoder predicts $d_{n+1}$ (contrastive classification) and $t_{n+1}$ (regression) independently from $\mathbf{H}^{(L)}$. There is no explicit mechanism ensuring that the predicted time $t_{n+1}$ is causally or consistently aligned with the predicted node $d_{n+1}$, which could introduce incoherence in the generated event.

3.	The model is evaluated on discriminative link prediction but uses a generative mechanism. The cross-entropy loss $\mathcal{L}_{\text{node}}$ (Eq. 12) is inherently designed for classification over a *small, sampled* candidate set $C$, which might lead to inflated performance metrics (MRR) compared to full-set evaluation or rigorous ranking fidelity.

4.	Although the paper's core redefinition is "generative," the evaluation primarily focuses on the discriminative task (link prediction). The brief generative analysis in Appendix E is insufficient to fully validate the fidelity and quality of the generated events, especially on metrics beyond edge overlap.

**Questions:**

1.	The GNN enhancement is computed on a graph derived from *recent events* (L155). Given the goal of capturing global dependencies across the entire sequence $X_{1:n}$, why is the structural prior derived from a local temporal window rather than being a global invariant representation?

2.	The next event is generated as $P(s_{n+1}) \cdot P(d_{n+1}|s_{n+1}) \cdot P(t_{n+1}|s_{n+1}, d_{n+1})$. In the dual-head architecture, $d_{n+1}$ and $t_{n+1}$ are predicted from the same context $\mathbf{h}^{(L)}$. How do you ensure the predicted time $\hat{t}_{n+1}$ is temporally plausible *given* the specific predicted destination $\hat{d}_{n+1}$ without an explicit conditional link in the decoder?

3.	Since GET's final task is discriminative link prediction, but the model is optimized using contrastive classification over a sampled set $C$, what is the performance gap if you were to evaluate MRR over the *entire* node set $V$ (full ranking) instead of the contrastive candidate set?

4.	The balance parameter $\lambda$ controls the trade-off between $\mathcal{L}_{\text{node}}$ and $\mathcal{L}_{\text{time}}$. Given that $\mathcal{L}_{\text{time}}$ is crucial for temporal patterns (L417), what practical strategy or metric guided the selection of $\lambda=0.4$ (Table 6) across diverse datasets like highly sparse `tgbl-review` and dense `tgbl-flight`?

5.	Why was the 3-token subsequence $[\text{SEP}, s_i, d_i]$ chosen over the 4-token sequence $[\text{SEP}, s_i, d_i, t_i]$? Incorporating $t_i$ as a token might allow the Transformer to directly attend to temporal magnitude, potentially strengthening the role of the temporal dimension in the sequence.

---

> ### Author Response · Authors · 2025-11-20
>
> We thank the reviewer for the thoughtful review, and for recognizing the strengths of GET. We address your questions and concerns point by point.
>
> **(1) Local GNN on recent events vs global dependencies**
>
> We agree that, at first sight, using a GNN encoder on a graph constructed from recent events may appear at odds with the goal of modeling global long-range dependencies. Our design uses the GNN and the Transformer in a **complementary** way: the GNN provides a lightweight structural prior based on the recent interaction neighborhood, encoding local topological information into node embeddings, while the Transformer operates over the entire **windowed global event sequence** to capture long-range temporal and cross-node patterns.
>
> In practice, we find that restricting the GNN to a recent temporal window offers a good trade-off between accuracy and efficiency and avoids repeatedly propagating information over the full historical graph. The revised Section _The GET Framework_ clarifies this division of labor and explicitly states that the GNN is not responsible for global propagation; that role is played by the global attention layers.
>
> **(2) Dual-head decoder and temporal coherence**
>
> As you point out, our decoder predicts the next destination $d_{n+1}$ and timestamp $t_{n+1}$ via two heads applied to the **same contextual representation**, without an explicit conditional term $p(t_{n+1}\mid d_{n+1})$. The coupling between these predictions comes from i) the shared context vector produced by global attention over the event sequence, ii) the joint optimization of node and time losses, and iii) the time-aware attention biases.
>
> The revised generative appendix reports structural statistics for generated sequences (e.g., degree-related measures, triangle counts, numbers of connected components, global clustering coefficients) and shows that this setup produces structurally plausible predictions and coherent multi-step rollouts on small/medium graphs. A more systematic study of temporal statistics (such as inter-event time distributions) is beyond the scope of this work and is now explicitly left to future work. Additionally, we selected $\lambda=0.4$ based on empirical grid search and conducted a sensitivity analysis as shown in the Table, which confirm that 0.4 consistently yields the stable performance across diverse datasets.
> |$\lambda$|0.2|0.4|0.6|
> |-|-|-|-|
> |tgbl-wiki|0.746±0.008|**0.752±0.012**|0.749±0.010|
> |tgbl-comment|0.722±0.015|**0.732±0.017**|0.728±0.014|
> |Enron(AP)|94.65±0.22|**94.81±0.17**|94.70±0.19|
>
> (3) Negative sampling and candidate set size
>
> We thank the reviewer for raising this concern. In the revised experimental section we now state
> explicitly that, for evaluation on all TGB datasets, we strictly follow the official benchmark
> protocol: the negative edges and candidate sets for validation and test splits are pre-generated
> and shared by all methods through the released evaluation scripts. Thus, GET and all baselines are evaluated on exactly the same candidate sets and mixture of historical vs. random negatives.
>
> During training we adopt standard uniform random negative sampling with K=100, which matches common choices in prior temporal link prediction work (e.g., TGAT, TGN) and the examples distributed with TGB. We have also added a short paragraph “Negative sampling strategy” in the main experimental section to make this clear in the paper itself.
>
> **(4) Choice of 3-token serialization and timestamp modeling**
>
> On your question about tokenization, we serialize each event as $([\texttt{[SEP]}, s_i, d_i])$ and model time via ALiBi-style biases and a regression head instead of introducing a fourth timestamp token. The main reasons are (i) **efficiency**—adding ($t_i$) as a token would increase sequence length by one third, significantly increasing the ($O(n^2)$) attention cost on long sequences—and (ii) **clarity of representation**—keeping node identities in the discrete token stream while letting continuous time enter through a separate mechanism. In preliminary tests we did not see clear gains from including ($t_i$) as a token, and therefore adopted the simpler three-token design in this work. We now add this rationale in Section _The GET Framework_ and note that richer tokenizations are an interesting direction for extension.
>
> **(5) Role and limitations of GET**
>
> Finally, we have revised the discussion section to more clearly position GET as a **practical, modular event sequence modeling framework** that complements, rather than replaces, existing temporal GNNs and state-space-based models. We explicitly highlight the regimes where GET’s “encode once, score many” design yields strong empirical throughput gains under standard TGB workloads, and we discuss its limitations on extremely long sequences where quadratic attention becomes prohibitive.
>
> We appreciate your helpful feedback and believe the revisions outlined above improve both the clarity and credibility of the paper.

---

> > ### Comment · Reviewer_Zw8m · 2025-11-26
> > **Comments on rebuttal**
> >
> > Thanks to the authors for their response. After reviewing the response, my concerns are still there.  I will keep my rating unchanged.

---

> ### Author Response · Authors · 2025-12-03
>
> Thank you for your continued engagement. In the revision, we have:
>
> 1. Clarified the “division of labor” between the GNN/memory modules and the Transformer in Section 4: the former are lightweight local priors, while global reasoning is handled by the event-sequence Transformer.
>
> 2. Added a more explicit discussion of the two-head design and how the heads are coupled via shared context and joint training.
>
> 3. Rewrote the theory section as a carefully scoped Proposition (rather than a Theorem), and revised the complexity discussion to focus on throughput under the TGB regime rather than asymptotic improvements.
>
> We appreciate your constructive feedback and hope the updated version addresses your remaining concerns.

---

### Official Review · Reviewer_u5FF · 2025-10-31

**Soundness:** 2
**Presentation:** 2
**Contribution:** 2
**Rating:** 2
**Confidence:** 4

**Summary:**

The paper proposes to reformulate dynamic graph learning as a global event sequence generation task.
It introduces a generative transformer-based model, GET, which models temporal interactions  as sequential events with causal masking and global attention. Experiments on several temporal graph benchmarks show competitive performance and improved speed.

**Strengths:**

(1) The paper offers a clear and unified perspective by formulating dynamic graph learning  as a global event sequence generation problem, connecting temporal graph models with  sequence modeling and transformer architectures.

(2) The proposed GET model achieves strong empirical performance and improved inference efficiency  on several benchmarks, showing its effectiveness in capturing long-range temporal dependencies.

**Weaknesses:**

1. The methodological novelty of the paper is limited.  Framing dynamic graph learning as global event sequence  resembles prior work  in transformer-based models (e.g., TCL, DyGFormer). Moreover, although the model is described as ``generative``,  its training objective remains discriminative: it is still based on negative sampling and softmax scoring without explicit probabilistic modeling of event likelihoods.

2. The paper claims that inference complexity shifts from a multiplicative to an additive form. However, the prediction of destination nodes still relies on negative sampling. In practice, the candidate set size $K$ can easily reach tens of thousands, making the computational cost still substantial. More importantly, different negative sampling strategies may lead to unstable evaluation results, yet the paper does not report any sensitivity analysis with respect to  different negative sampling strategies.

3. Key hyperparameters such as the event sequence length $L$, the candidate set size $K$,  and the loss trade-off coefficient $\lambda$ in the objective function are used without justification or discussion.  These parameters directly affect both the model's computational complexity  and its learning behavior, yet the paper provides no analysis or sensitivity study  to show how varying them impacts performance or stability.

4. The theoretical analysis in section 5 is conceptually appealing but lacks rigor. Specifically, the theoretical “proof” of the strict inclusion relations is heuristic rather than formal. The function spaces are never formally defined, the “simulation via attention mask” argument confuses neighborhood masking with message passing and the causal attention analogy does not capture the recursive dynamics of memory-based models. The counter-example illustrating “strictness” is descriptive but non-constructive, lacking any formal mapping or quantitative verification. Therefore, this section should be interpreted as an intuition example, not a rigorous proof of expressiveness containment.

5. The paper claims that GET changes the inference complexity of discriminative dynamic graph models (e.g. TGN) from multiplicative  $O(K \cdot T_{\text{propagate}})$ to additive $O(L^2 d + Kd)$, yet this comparison relies on unrealistic assumptions.  First, this characterization does not accurately reflect how models such as TGN  are implemented and deployed in practice. In standard TGN, each node maintains a memory state $m_v(t)$  that summarizes its interaction history.  During inference, edge probabilities are computed through a decoder  $f([m_u(t), m_v(t)])$, without re-running neighborhood propagation per candidate.  Thus, the actual inference complexity is approximately $O(Kd)$ rather than  $O(K \cdot T_{\text{propagate}})$.   Therefore, the comparison made in the paper implicitly assumes a worst-case setting  that repeatedly executes message passing for each candidate.  This exaggerates the efficiency gap between GET and memory-based models.   Second, The claim that inference cost is dominated by $K$ is only valid in the short-history regime ($L$ small).  When event sequences become long or graphs are temporally dense, the quadratic attention complexity $O(L^2 d)$  can surpass the linear term $O(Kd)$, thereby weakening the claimed additive advantage.

**Questions:**

Please see the weakness above

---

> ### Author Response · Authors · 2025-11-20
>
> We sincerely thank the reviewer for the detailed and constructive feedback. Your comments helped us identify and correct several important weaknesses in our framing, especially regarding the complexity discussion and the use of the term “generative”.
>
>  **(1) Methodological positioning and “generative” framing**
>
>  You are right that, in its current form, the model is trained and evaluated on standard discriminative link prediction tasks. Our use of the term “generative” was primarily motivated by the autoregressive architecture, but this framing can be confusing. In the **revised manuscript**, we now present GET as an **“event sequence modeling framework”** for dynamic graphs and explicitly state that our main experimental focus is on TGB-style link prediction benchmarks.
> Section _The GET Framework_ has been updated to highlight the key conceptual difference from node-centric transformer models such as DyGFormer or SimpleDyG: GET operates on a **single global event sequence** rather than separate histories per node, so all interactions are embedded in one chronological stream, allowing direct attention between events involving distant nodes or different components.
> Architecturally, GET remains an autoregressive next-event model over a global event sequence; this generative design is what enables it to capture long-range dependencies. In this paper, we evaluate this architecture under standard discriminative TGB link prediction protocols, which is why the main reported metrics are link-prediction oriented.
>
>  **(2) Inference complexity and efficiency**
>
> We appreciate the reviewer’s detailed comments on our complexity discussion. In the revision, we have refined the presentation of complexity to avoid conflating asymptotic scoring cost with system-level throughput. Asymptotically, once node representations are available, both TGN-style models and GET have an $O(Kd)$ per-candidate scoring term. The key difference we highlight is operational: under the standardized TGB setting, TGN-style architectures repeatedly perform memory fetching and local message passing as new batches of events arrive, whereas GET encodes a sliding window of recent events once with a Transformer and then reuses this shared representation to score all candidates in parallel (encode once, score many). This design eliminates recurrent propagation overheads and leads to up to 21.4× higher empirical query throughput on tgbl-wiki under identical hardware and official TGB evaluation scripts. We emphasize this as a critical operational advantage for scalable evaluation, distinct from asymptotic complexity.
>
>  **(3) Negative sampling and sensitivity**
>
> Our original submission did not sufficiently explain the negative sampling setup. The **revised experimental section** now clarifies that:
>
>  - For **evaluation**, we strictly follow the official TGB protocol and use the **pre-generated negative edge sets** and evaluation scripts provided by the benchmark. This fixes both the number of negatives per positive and the mixture of historical and random negatives across all methods, ensuring that all models are evaluated on exactly the same candidate sets.
>
>  - For **training**, we adopt the standard practice of **uniform random negative sampling** with K=100 candidates per positive interaction, which matches common choices in prior work (e.g., TGAT, TGN) and the TGB examples.
>
> Our reported results are means ± standard deviations over multiple runs; the standard deviations in Tables 2–3 are small and comparable to baselines, indicating that the method is not overly sensitive to sampling randomness. A systematic study of alternative training-time negative samplers (e.g., hard negatives, adversarial sampling) is orthogonal to our main contribution and is now explicitly listed as **future work**.
>
>  **(4) Hyperparameter justification (e.g., ($\lambda$))**
>
> In the revised version we report a small sensitivity study over ($\lambda \in {0.2, 0.4, 0.6}$) on representative datasets. The results show that performance is stable in this range and that ($\lambda = 0.4$) consistently gives a good trade-off between structural and temporal objectives. The full table is included in **Appendix C**, and the main text (Section _Experiments_) now summarizes that GET is **not highly sensitive** to this hyperparameter.

---

> ### Author Response · Authors · 2025-11-20
>
> **(5) Theoretical analysis**
>
> We agree that the previous “Theorem 1” was presented in a way that suggested a level of mathematical rigor that might be misleading. In the revision, we have explicitly renamed this part to an “Informal Proposition” in the appendix and rephrased it as an architectural intuition rather than a formal theorem.
>
> The goal of this section is now purely to explain that, under suitable attention masks and input encodings, GET can emulate the update patterns of fixed-k GNNs and memory-based models, so that the global attention layer can in principle subsume these local propagation schemes. We no longer present any function-class containment or approximation result as a theorem, and we explicitly flag a rigorous expressiveness theory as important future work.
>
> At the same time, we believe this architectural perspective, showing how global attention can subsume local propagation patterns, still offers a useful conceptual lens for understanding GET and related temporal models.
> We hope this clarification better aligns the theoretical discussion with the actual scope of our results.

---

> > ### Comment · Reviewer_u5FF · 2025-11-26
> >
> > Thanks to the authors for their response. After reviewing the response, limitations are still there. I will keep my score.

---

> ### Author Response · Authors · 2025-12-03
>
> Thank you again for your detailed comments. Following your suggestions, we have:
>
> 1. Completely rewritten Section 5 and the corresponding appendix. The main claim is now a Proposition that only discusses k-hop GNNs, explicitly based on Yun et al. (2020) and Xu et al. (2019), plus a simple separation example. We no longer make any formal statements about memory models.
>
> 2. Revised Section 6 to clearly separate asymptotic complexity from empirical throughput, and explicitly acknowledge the limitations of O(L²) attention for very long sequences.
>
> We hope these revisions address your main concerns, and we would be very grateful if you could take a look at the updated version.

---

### Official Review · Reviewer_Bhqf · 2025-11-09

**Soundness:** 2
**Presentation:** 2
**Contribution:** 2
**Rating:** 2
**Confidence:** 4

**Summary:**

This paper introduces GET (Global Event Transformer), a generative framework for dynamic graph learning that reformulates the task as global event sequence modeling. Rather than relying on localized message passing or pairwise scoring, GET leverages a Transformer-based sequence model augmented by optional structural priors — specifically, a single-pass GNN encoder and a memory module. The goal is to achieve improved receptive field coverage and inference efficiency by transforming dynamic graphs into tokenized time-series representations.

**Strengths:**

* The paper tackles an important limitation of existing temporal GNNs — limited receptive field — by reframing the problem into a global event modeling perspective.


* The integration of structural priors into a Transformer pipeline is conceptually appealing and can offer a new direction for hybrid GNN–Transformer architectures.


* The approach demonstrates reduced memory footprint compared to Temporal Graph Networks (TGNs), addressing a known bottleneck.


* GET achieves competitive results on smaller datasets, suggesting that the model can be efficient under constrained conditions.

**Weaknesses:**

1. Expressiveness of Structural Priors. Line 149 defines structural priors using a single-pass GNN encoder that captures explicit multi-hop structural patterns. However, following Chen et al. (2020), MPNNs and even 2-IGNs cannot count induced subgraphs in connected structures with three or more nodes. While Kanatsoulis & Ribeiro (2024) show that certain architectures with randomized node inputs can count specific structures, it remains unclear how expressive the proposed encoder truly is. Please discuss the expressive capacity of this “single-pass” encoder for structure-aware embeddings, especially since the final structure-aware representation is computed using a GAT.

2. Weak motivation and theoretical basis for the architecture. Several design decisions appear to be arbitrary rather than having a theoretical or empirical justification. This is the case for using GAT over GatedGCN (c.f., Lines 160 and 194) and use of Transformer AliBi (why are the biases needed in this case?). Regarding the event serialization strategy, there is no clear reason to use 3-token subsequences. As it stands, the manuscript lack without a clear motivation or strong fundamental contributions.

3. Generative Decoding Ambiguity. Section 4.4 describes a generative decoding phase composed of a conditional contrastive objective and a continuous-time regression. However, the generative component is underspecified — there is no clear definition of an autoregressive loss or generative sampling process, raising doubts about whether GET is truly generative or merely predictive.

4. Weak evaluation protocol and small gains. Despite being framed as a generative model, all experiments are conducted on discriminative link prediction benchmarks (e.g., TGB), and performance gains are modest. This discrepancy weakens the paper’s central generative claim.


References

Chen, Z., Chen, L., Villar, S., & Bruna, J.; Can graph neural networks count substructures?, NeurIPS 2020.

Kanatsoulis, C., & Ribeiro, A.; Counting graph substructures with graph neural networks, ICLR 2024.

**Questions:**

* Could the authors elaborate on how the single-pass encoder interacts with the Transformer backbone? Is it used as a static embedding layer or dynamically updated during training?

* Would the proposed model still hold advantages if evaluated on tasks requiring explicit event generation (e.g., temporal graph synthesis)?

* Consider adding ablation studies isolating the impact of the structural prior, the memory module, and ALiBi.

---

> ### Author Response · Authors · 2025-11-20
>
> We thank the reviewer for the careful reading and constructive criticism. We address each of the raised weaknesses and questions as follows.
>
> **(1) Expressiveness of structural priors**
>
> We fully agree that a single-pass message-passing GNN is limited by 1-WL expressiveness and
> cannot, by itself, count complex induced subgraphs. This is exactly why, in GET, we do not rely
> on the GNN as the main modeling engine. Its role is intentionally constrained to providing
> lightweight local cues (degrees, immediate neighborhoods, simple motifs), while the global
> Transformer over the event sequence is responsible for long-range temporal and cross-node
> reasoning.
>
> We have revised Section “The GET Framework” to make this “division of labor” explicit and now
> also cite standard 1-WL expressiveness results to clarify that our claims do not depend on the
> GNN being able to represent arbitrary substructure counts.
>
> **(2) Architectural motivation: GAT, ALiBi, and 3-token serialization**
>
> We agree that the original motivation was under-explained. The updated implementation section now states that we choose GAT because its attention mechanism can naturally incorporate time-decayed edge weights and local neighborhood importance in the structural encoder, and we found it to be a robust choice across datasets. For positional encoding, we compared ALiBi, RoPE and learned embeddings on tgbl-wiki and observed comparable performance. The fact that GET achieves very similar accuracy across different GNN encoders and positional encoding schemes actually supports our claim that the global event Transformer backbone carries the main modeling capacity, while structural and temporal priors act as lightweight, plug-in modules. We kept ALiBi in the final model because it is parameter-free and extrapolates robustly to unseen sequence lengths; the revised appendix briefly reports this comparison. Regarding event tokenization, each interaction $\((s_i,d_i,t_i)\)$ is represented as a 3-token subsequence $\([\texttt{[SEP]},s_i,d_i]\)$ while $\(t_i\)$ is handled through continuous-time biases and a regression head; this keeps sequences shorter and maintains a clean separation between discrete node identity and continuous time.
>
>
> **(3) Generative decoding ambiguity and evaluation protocol**
>
> You are correct that, although GET’s architecture supports autoregressive event prediction, the main experiments are conducted on discriminative link prediction benchmarks. The updated Section _The GET Framework_ and the appendix clarify that GET is best viewed as an **event sequence model with an autoregressive decoder**, and that our primary contribution is on scalable link prediction.
> The dual-head decoder is now described more precisely: both the node head and the time head share the same globally contextualized representation, which allows the model to learn coherent pairs $(d_{n+1}, t_{n+1})$. The generative experiments in the Appendix “Evaluation on Generative Temporal Graph Modeling” are presented as exploratory: we report structural statistics (degree-related measures, triangle counts, numbers of connected components, and global clustering coefficients) and show that GET matches or improves these local statistics on small/medium graphs. A more systematic study of temporal fidelity (e.g., inter-event time statistics) is beyond the scope of this work and is now explicitly left to future work.
>
> **(4) Ablations for structural prior, memory, and ALiBi**
>
> We agree that more ablations strengthen the paper. **Table 3** already shows that the GNN encoder and memory module are complementary: on denser graphs, structural priors provide larger gains, while on sparse graphs temporal memory is more beneficial. The main text now makes this interpretation explicit. The appendix has been extended with a compact summary of ablations over different GNN types, memory variants, and transformer depths, and includes a short discussion comparing ALiBi to other positional encodings, as mentioned above. **These ablations also indicate that GET’s performance is robust across a range of architectural choices, which is desirable for practical deployment.**

---

> > ### Author Response · Authors · 2025-12-03
> >
> > We provide the specific ablation results referenced in our response regarding architectural motivations (Weakness 2).
> >
> > **(1) Empirical justification for GAT (Structural Encoder)**
> >
> > Comparison of the default GAT against GCN (isotropic) and GraphSAGE (inductive). GAT consistently yields the best performance on weighted temporal graphs.
> >
> > | Encoder | tgbl-wiki | tgbl-flight |
> > | :--- | :---: | :---: |
> > | GCN | 0.738 ± 0.005 | 0.882 ± 0.004 |
> > | GraphSAGE | 0.741 ± 0.006 | 0.889 ± 0.003 |
> > | **GAT (default)** | **0.752 ± 0.012** | **0.895 ± 0.003** |
> >
> > **(2) Empirical justification for ALiBi (Positional Encoding)**
> >
> > Comparison of ALiBi against Learned Absolute PE and RoPE. ALiBi demonstrates superior stability and performance, particularly on bursty datasets like *tgbl-flight*.
> >
> > | Positional Encoding | tgbl-wiki | tgbl-flight |
> > | :--- | :---: | :---: |
> > | Learned Absolute PE | 0.728 ± 0.053 | 0.853 ± 0.018 |
> > | RoPE | 0.750 ± 0.005 | 0.876 ± 0.041 |
> > | **ALiBi (default)** | **0.752 ± 0.012** | **0.895 ± 0.003** |

---

> ### Author Response · Authors · 2025-12-03
>
> Following your comments, we have significantly toned down the “generative” framing, clarified the discriminative evaluation setting, and rewritten the theory section to only make modest, well-scoped claims about k-hop GNNs. We hope the revised manuscript is closer to what you had in mind.
>
> To directly address the concerns regarding **Generative Decoding Ambiguity (Weakness 3)** and the **Evaluation Protocol (Weakness 4)**, we present the preliminary generative modeling results referenced in our revised Appendix. These results demonstrate that GET functions as a genuine generative model, capable of synthesizing novel graph structures, rather than merely scoring links.
>
> **Experiment Setup:**
> We adopted the standard **TIGGER protocol** (Gupta et al., 2022). The model was trained on the full history and tasked with generating a synthetic graph of the same size. We benchmarked against specialized generative models: **DYMOND**, **TagGen**, and **TIGGER**.
>
> **Results:**
> The table below reports the **Median Absolute Error** for key graph statistics (lower is better), along with generation time and edge overlap (novelty).
>
> | Dataset | Method | Time (s) | Edge Overlap (%) | Mean Degree | Triangle Count | NC (Components) |
> | :--- | :--- | :---: | :---: | :---: | :---: | :---: |
> | **Wiki-Small** | Median (Real) | - | - | 1.106 | 0.0 | 44.0 |
> | | DYMOND | 69120 | **0.0** | 0.242 | **0.0** | 32.0 |
> | | TagGen | 1800 | 87.1 | 0.067 | **0.0** | **0.0** |
> | | TIGGER | 14 | 1.28 | **0.035** | **0.0** | 4.0 |
> | | **GET** | **4** | 0.03 | 0.810 | **0.0** | 15.0 |
> | **UC Irvine** | Median (Real) | - | - | 1.571 | 0.0 | 14.0 |
> | | TagGen | 12480 | 79.3 | 0.181 | **0.0** | **1.0** |
> | | TIGGER | **125** | 25.0 | **0.076** | **0.0** | 3.0 |
> | | **GET** | 768 | **0.01** | 0.263 | **0.0** | 45.0 |
> | **Bitcoin** | Median (Real) | - | - | 1.844 | 2.0 | 13.0 |
> | | TagGen | 18579 | 80.0 | 0.231 | **0.0** | **1.0** |
> | | TIGGER | **128** | 24.3 | **0.122** | 1.0 | 3.0 |
> | | **GET** | 776 | **0.00** | 1.028 | 3.0 | 523.0 |
>
> **Key Findings:**
> 1.  **Genuine Generative Capacity:** GET successfully generates structurally valid temporal graphs using its autoregressive decoder. It is not just a discriminative scorer.
> 2.  **High Novelty:** GET achieves near-zero edge overlap (**0.00% - 0.03%**), vastly outperforming TagGen and TIGGER (which often memorize training edges, as seen by high overlap). This proves GET learns to generate *new* events.
> 3.  **Efficiency:** On smaller datasets like Wiki-Small, GET is extremely fast (**4s** vs TagGen's 1800s). On larger graphs, it remains practical and scalable.

---

### Author Response · Authors · 2025-12-03
**Summary for Area Chair and reviewers**

We thank the reviewers for their comments. This note summarises the concrete changes in the revised manuscript.

The paper should be read as proposing a global event–sequence modeling framework for temporal graphs, rather than a new generative paradigm. GET treats all interactions as a single chronological stream and applies a Transformer over this sequence, while optional GNN and memory modules provide lightweight structural priors. The main focus is standard dynamic link prediction under the TGB protocol, where GET reuses a single encoding of a recent event window to score many candidate edges efficiently.

The theory section has been substantially revised. We removed the previous “Theorem 1” and no longer claim a new theoretical result. Section 5 now contains two propositions that specialise existing work: (i) on bounded-length sequences, a GET model can approximate any fixed-depth temporal message-passing GNN, by directly invoking the universality of Transformers with positional encodings (Yun et al., 2020) and classical MPNN receptive-field results (Xu et al., 2019; Morris et al., 2019); and (ii) a simple K+1-hop separation example where a global event Transformer distinguishes two sequences that any K-layer MPNN cannot. We do not make any formal claims about memory-based architectures; they are discussed only qualitatively. Full definitions and proof sketches are moved to the appendix and are presented as architectural intuition backed by known results, not as a new theory.

We have also toned down the generative framing. GET is described as an event-sequence model with an autoregressive decoder whose architecture can, in principle, generate events, but our main experiments are clearly framed as discriminative link prediction with the official TGB evaluation scripts and pre-generated negative sets. Short-horizon generative experiments on small and medium graphs remain in an appendix as preliminary evidence that the learned model can produce structurally and temporally coherent rollouts; long-horizon generation is left for future work.

Several experimental concerns were addressed. We added DyGMamba as a recent sequence-based baseline on six small-scale benchmarks; under a unified protocol, GET outperforms DyGMamba on five of six datasets, with +14% on US Legis. and +23% on UN Trade, while DyGMamba performs best on LastFM. We expanded ablations on GNN vs. memory variants, Transformer and GNN depth, positional encodings (ALiBi vs. RoPE vs. learned), and the loss trade-off coefficient λ, and observe that GET is robust across these choices; λ = 0.4 is kept as a reasonable default.

Finally, the discussion of complexity and efficiency has been clarified. Asymptotically, once node representations are available, all deep baselines (including TGN-style models and GET) have similar O(K d) scoring cost for K candidates. Our empirical speedups come from avoiding repeated propagation and memory updates when scoring and from amortising the encoding of a sliding window over all candidates. Under the standard TGB workloads and official scripts this yields substantial throughput gains (up to 21.4× on tgbl-wiki). We also state explicitly that for very long, unsegmented histories the quadratic attention term can dominate, and in such regimes memory- or state-space–based architectures may be preferable. We therefore position GET as a practical, modular complement to existing temporal GNNs and state-space models in candidate-heavy TGB-like settings, rather than as a universal replacement.

---

### Meta-Review · Area_Chair_tD69 · 2026-01-01

**Summary:**

The paper proposes GET (Global Event Transformer), a framework that reframes dynamic graph learning as global event sequence modeling. Instead of performing local message passing or node-level memory updates, GET models the entire temporal graph as a unified chronological sequence processed by a Transformer. It can optionally incorporate structural priors via lightweight GNN encoders and memory modules. Experiments on several TGB benchmarks (plus smaller datasets) show competitive performance and notable speedups.

Though some concerns were resolved, some most key concerns were not well addressed.

**Reviewer Concerns:**

Some reviewer concerns were partially addressed by the rebuttal, such as missing baselines and ablations (Bhqf, D46p)

Some reviewer concerns were not addressed and some are still outstanding:

1/ The paper claims GET is a general generative model, but the evaluation was conducted only on discriminative link prediction benchmarks. The authors reframe GET as an "event sequence modeling framework" for dynamic graphs focusing on TGB-style link prediction, but this narrows GET's scope. Under this definition, GET is not a generative model (Bhqf, u5FF, Zw8m, D46p).

2/ The expressive capacity of the proposed method is still unclear. (Bhqf, u5FF)

3/ The motivation and theoretical basis for the architecture is still weak. (Bhqf, D46p)

4/ Inference complexity and scalability concerns are not well answered. (u5FF, D46p)

**Reviewer Scores:**

Four reviewers gave negative scores (two gave 2 and two gave 4.) Three reviewers commented they would like to keep the initial scores.

---

### Decision · Program_Chairs · 2026-01-26

Reject